# Ribosome biogenesis as a potential therapeutic target in KRAS mutant colorectal cancer

Yui Tanaka[1,2,11], Mizuho Sakahara[1,11], Hitomi Yamanaka[1], Yasuko Natsume[1], Daisuke Kusama[1], Kohei Kumegawa [3], Harunori Yoshikawa [4], Yuich Abe[5,10], Koji Okabayashi[2], Shimpei Matui[2], Yuko Kitagawa[2], Naohiko Koshikawa[6], Hiroki Osumi [7], Eiji Shinozaki[7], Satoshi Nagayama [8], Jun Adachi [5], Reo Maruyama [3,9] & Ryoji Yao [1] ✉

Molecular targeted therapies targeting KRAS signaling have significantly improved patient outcomes, but they have not achieved sufficient therapeutic efficacy in colorectal cancer (CRC). Here, we demonstrate that a subset of KRAS-mutant CRC cells transitions to a cellular state characterized by enhanced ribosome biogenesis upon KRAS signaling inhibition. The mitogen-activated protein kinase kinase inhibitor, trametinib, and AMG510 induce a cellular state characterized by a gene expression profile highly enriched for ribosome biogenesis. We find that they are vulnerable to the inhibition of RNA polymerase I, and they exhibit synergistic anti-tumor effects with trametinib in an autochthonous mouse model of intestinal tumors and human patient-derived organoids (PDOs). These observations demonstrate that high ribosome biogenesis induced by KRAS inhibition is indispensable to maintain this cellular state and is a potential therapeutic target. Overall, this study reveals novel mechanisms of drug tolerance to KRAS inhibition, thereby facilitating the development of new therapeutic strategies.

Colorectal cancer (CRC) is among the most commonly diagnosed cancers and the third most common cause of cancer-related deaths worldwide. Owing to the high prevalence of *KRAS*-activating mutations, the RAS pathway is considered a promising target for CRC treatment[1]. Specific inhibitors of RAS pathway molecules and mutant allele-specific inhibitors have been successfully developed[2–5]. However, resistance inevitably occurs, and the insufficient efficacy of these agents in trials limits their use in clinical practice as monotherapies.

Reactivation of signaling and pathway bypass confer therapeutic resistance to several types of cancers[6–8]. In CRC, combinational inhibition of the RAS pathway and epidermal growth factor receptor has been tested in *RAS*-mutant CRC. However, despite the improvement in patient prognosis using combination chemotherapies, their efficacy is insufficient, warranting further improvement for CRC treatment[3,9,10].

Tumor tissues are hierarchically organized with heterogeneous cells, and their plasticity is a possible cause of drug resistance[11,12].

[1]Department of Cell Biology, Cancer Institute, Japanese Foundation for Cancer Research, Tokyo, Japan. [2]Department of Surgery, Keio University School of Medicine, Tokyo, Japan. [3]Cancer Cell Diversity Project, NEXT-Ganken Program, Japanese Foundation for Cancer Research, Tokyo, Japan. [4]Fujii Memorial Institute of Medical Sciences, Institute of Advanced Medical Sciences, Tokushima University, Tokushima, Japan. [5]Laboratory of Proteomics for Drug Discovery, Center for Drug Design Research, National Institute of Biomedical Innovation, Health and Nutrition, Osaka, Japan. [6]Department of Life Science and Technology, Institute of Science Tokyo, Yokohama, Japan. [7]Department of Gastrointestinal Oncology, Cancer Institute Hospital, Japanese Foundation for Cancer Research, Tokyo, Japan. [8]Department of Surgery, Uji-Tokushukai Medical Center, Uji, Japan. [9]Project for Cancer Epigenomics, Cancer Institute, Japanese Foundation for Cancer Research, Tokyo, Japan. [10]Present address: Immunoproteomics Laboratory, Institute for Glyco-core Research (iGCORE), Gifu University, Gifu, Japan. [11]These authors contributed equally: Yui Tanaka, Mizuho Sakahara. ✉e-mail: ryao@jfcr.or.jp

Cancer stem cells are characterized by their self-renewal and differentiation capacities, and cell stemness is a key feature responsible for the drug resistance of multiple cancer types[13]. Initially, cancer stem cells were thought to be slow-replicating cells that were dormant in several cancer types, which contributed to their drug resistance[14]. However, their proliferative properties vary depending on the cancer type, and low levels of tumor dormancy have been reported in CRC cells[15]. Cancer stem cells have unique cellular properties, such as cellular metabolism, energy consumption, and epithelial-mesenchymal transition[16]. Ribosome biogenesis determines the stemness of embryonic and adult stem cells[17]. In line with this, transcription of ribosomal DNA defines the stem cell hierarchy in CRC[18]. Selective pressures imposed by chemotherapeutic agents eliminate the non-cancer stem cells, leading to an enrichment of cancer stem cells[19,20]. Therefore, specific characteristics of the cellular states and vulnerabilities of drug-resistant cells need to be elucidated to enhance the efficacy of chemotherapeutic agents. Here, we show KRAS inhibition induces a cellular state marked by ribosome biogenesis, which is vulnerable to RNA polymerase I inhibition.

## Results

### Mitogen-activated protein kinase kinase (MEK) inhibition promotes a cellular state with elevated ribosomal biosynthesis and reduced protein synthesis resembling the ground state of ESCs

We previously showed that 75% of KRAS-mutated CRC PDOs are resistant to the MEK inhibitor, trametinib[21]. To elucidate the molecular basis of this resistance, we analyzed a previously characterized trametinib-resistant PDO (HCT24-8) carrying a KRAS G12A mutation. The PDOs was treated with 10 nM trametinib for 24 h or left them untreated and obtained the multiplexed single-cell expression profiles. Cells were divided into five groups via unbiased clustering and visualized using uniform manifold approximation and projection (UMAP) (UMAP; Fig. 1a, left panel). Each cluster was initially annotated by calculating the $p$-value of overlaps using Fisher's exact test between the differentially expressed genes (DEGs) of each cluster and CRC markers reported previously[22]. Two absorptive cell clusters, C2 and C3, were further characterized using a single-cell transcriptomic atlas (Supplementary Fig. 1b)[23]. Notably, trametinib treatment increased the proportion of C1 cells (Supplementary Fig. 1a), which were annotated as intestinal crypt stem cells of the colon, and decreased the proportion of C2 cells, which were enriched in mature enterocyte markers (Fig. 1b). C3 and C4 were annotated as absorptive cells and transit-amplifying cells, respectively, with their proportions remaining relatively unchanged. C5 was annotated as goblet cells, which were absent under untreated conditions but was induced following trametinib treatment.

Gene set enrichment analysis of the Gene Ontology cellular components (GOCC) revealed that C1 cells were enriched in multiple ribosome-related signatures (Fig. 1c), including ribosome gene sets (normalized enrichment score [NES] = 8.41, $p$adj = 0.1; Fig. 1c and Supplementary Fig. 1c). Further analysis of hallmark genes in the molecular signatures database (MsigDB) revealed Myc target V1 as the most enriched gene set (NES = 3.84, $p$adj = 0.04; Fig. 1d and Supplementary Fig. 1d). As Myc plays a critical role in the coordinated transcription of ribosome genes in various types of stem cells[17,24], these observations suggest that cells in C1 exhibit elevated Myc-dependent transcription of ribosome genes. Protein levels of three ribosome components, PABP1, RPL13A and RPL23, identified through enrichment analysis were upregulated following trametinib treatment by $1.7 \pm 0.2$, $1.5 \pm 0.09$ and $1.4 \pm 0.1$ fold (mean ± SEM, $N = 3$ biological replicates), respectively, thereby confirming the transcriptome analysis (Fig. 1e).

Embryonic stem cells (ESCs) are pluripotent stem cells derived from epiblasts, and two phases of cell state have been defined[25]. Ground state represents cells that produce a fully unrestricted population, whereas the primed state exhibits a restricted pluripotency configuration (Fig. 1f). The cellular properties of ground state ESCs are maintained by Myc-dependent ribosome biogenesis[26]. Notably, MEK inhibition promotes the reprogramming of mouse ESCs from the primed state to the ground state[27]. As trametinib promoted the cell state with elevated ribosome biogenesis and Myc-dependent transcription in CRC PDOs, we compared their transcriptome profile with that of ground-state mouse ESCs. The gene expression profile of C1 was highly enriched in the ground state mouse ESC signature (NES = 4.12, $p$adj <0.001; Fig. 1f, lower panel and Supplementary Fig. 1c)[28]. Next, we compared the expression profiles of trametinib-treated CRC cells with those of two states of mouse epiblasts, as the pluripotent naïve ground state and primed state of ESCs represent the pre- and post-implantation epiblasts, respectively (Fig. 1g, upper panel). This analysis revealed that the expression profile of C1 was positively (NES = 2.17, $p$adj = 0.017) and negatively NES = −0.92, $p$adj = 0.97) enriched in pre-implanted ground and post-implanted primed epiblasts, respectively (Fig. 1g, lower panes, and Supplementary Fig. 1d)[29].

We next investigate the expression profile of the genes enriched in the ground state ESCs (Fig. 1f) and epiblast (Fig. 1g). Ten genes representing higher expression levels in ground state mouse epiblast were significantly highly expressed in C1 (adj-$p$-value < 0.05) (Fig. 1h and Supplementary Table 1). Interestingly, seven genes were also highly expressed in C5. Given that the proportion of C1 and C5 was increased by trametinib treatment, these results indicate that trametinib induces a cell population with the expression profile similar to the ground state mouse epiblast in CRC organoids. Elevated ribosome biogenesis in C1 were confirmed by calculating the gene score of ribosome biogenesis feature (Supplementary Fig. 1e).

To further evaluate the trametinib-induced cell state of CRC organoids, we analyzed the expression profile of human ESCs. The expression levels of ribosome subunits vary among distinct tissues, and some of them are highly expressed in ESCs[30]. We found RPS13A, RPS18, RPL7A and RPL36A, which are highly expressed in human ESCs and decreased upon differentiation, were highly expressed in C1 (adj-$p$-value < 1e10$^{-62}$) (Fig. 1i and Supplementary Table 2)[31]. RPL13A is known as a stem cell marker in human ESCs and was highly expressed in C1 and the trametinib-treated fraction of C3[32]. These observations highlight the shared expression profiles of ribosome-related genes between CRC organoids and human ESCs.

Low global translation rates, despite high levels of ribosome biogenesis, have been well-documented in pluripotent naïve ground-state ESCs[33]. Here, we evaluated these rates via immunofluorescence analysis. Fibrillarin, a methyltransferase for ribosome RNA processing in nucleoli that is necessary for the maintenance of pluripotency, is highly expressed in pluripotent ES cells[34]. We found that trametinib significantly elevated the fibrillarin content in CRC organoids ($p = 5.46$E-5; Fig. 1j). Moreover, trametinib significantly reduced the protein synthesis rate in CRC organoids, as measured via O-propargyl puromycin (OPP) incorporation ($p = 5.73$E-5; Fig. 1k)[35]. 5-ethylnyl-2'-deoxyuridine (EdU) incorporation was partially reduced by trametinib treatment, resembling pluripotent ESCs with remarkable global suppression of DNA replication ($p = 4.54$E-3; Fig. 1l)[36]. Collectively, these observations suggest that trametinib induces a cellular state in KRAS-mutant CRC PDOs in which Myc-dependent ribosome biogenesis is upregulated.

### Trametinib reduces the phosphorylation of RNA-binding molecules

Next, we focused on the phosphorylation profile of PDOs to systematically dissect the network that controls the cell state of CRC cells. We first compared the KRAS-mutant PDO, HCT24-8, with the isogenic KRAS wild-type PDO, HCT24-14 (Fig. 2a). Trametinib treatment significantly reduced the viability of HCT24-14 cells, whereas HCT24-8

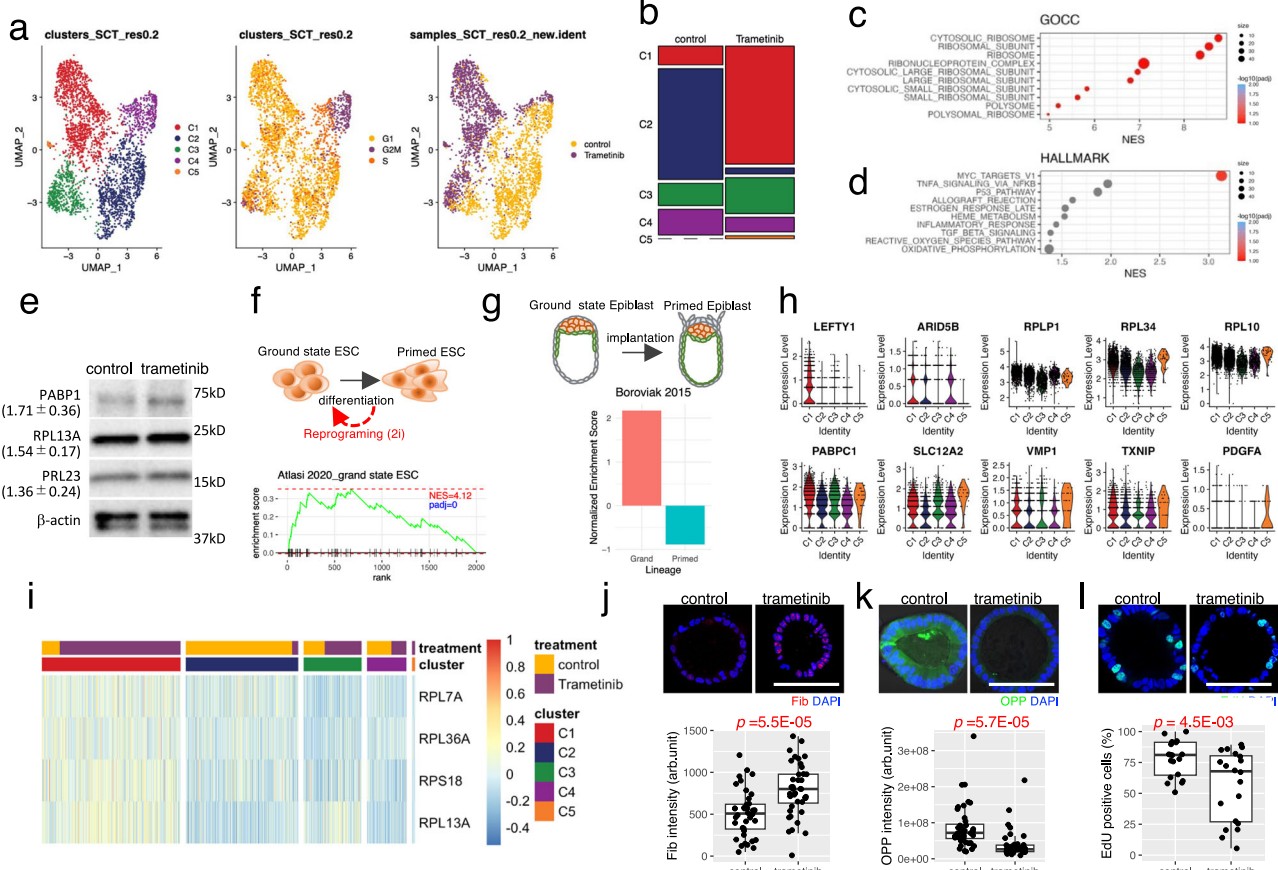

**Fig. 1 | Trametinib induces a cellular state resembling the ground state of embryonic stem cells (ESCs). a** Uniform manifold approximation and projection (UMAP) of patient-derived organoids (PDOs) carrying a *KRAS* G12A mutation. Single PDOs (HCT24-8) derived from a trametinib-resistant culture were treated with trametinib or left untreated, and multiplexed single-cell expression profiling was performed. Cells are colored by clusters (left), cell cycle score (middle), and treatment (right). A total of 1554 cells from the control group and 1657 cells from the trametinib-treated group were analyzed. **b** Proportion of cells in each cluster of untreated (control) and trametinib-treated (trametinib) PDOs. Enrichment analysis of C2 using Gene Ontology cellular component (GOCC) (**c**) and Hallmark (**d**). Statistical significance was evaluated using permutation testing (*n* = 1000). *P*-values were adjusted for multiple comparisons using the false discovery rate. **e** Immunoblot analysis of ribosomal proteins. Protein levels of indicated proteins in trametinib-treated (trametinib) and untreated (control) PDOs were evaluated. β-actin was shown as the loading control. *n* = 3 biological replicates. Data are

presented as fold-change mean ± SD (shown in parentheses). Gene set enrichment analysis of C2 with ground-state ESCs (**f**)[28] and epiblast (**g**)[29] are shown in the upper panels. **h** Violin plot depicting the expression of ground state genes in each cluster in CRC organoids. The expression of gene sets identified in the enrichment analysis of ground state ESCs (**f**) and epiblasts (**g**) were shown. Statistical analyses are provided in Supplementary Table 1. **i** Heatmap of genes known to have high expression levels in ESC. Statistical analyses are provided in Supplementary Table 2. Altered fibrillarin expression (**j**), global protein synthesis (**k**), and EdU incorporation (**l**) in trametinib-treated PDOs. Representative images are shown in the upper panels. Data of 40 PDOs (**j**, **k**) or 20 PDOs (**l**), obtained from four independent experiments are presented. Bar = 100 μm. The black line within the boxplot indicates the median. Bottom and top of the boxplot represent the 25th and 75th percentiles, respectively. Whiskers extend to 1.5 × IQR. *P* values were calculated using a two-sided Welch's two-sample t-test. Source data are provided as a Source data file.

cells were resistant, confirming the role of KRAS mutations in trametinib response (Fig. 2b)[21]. Using mass spectrometry-based phosphoproteomic analysis optimized for Matrigel-embedded organoids[37], we identified 3043 proteins and 8354 phosphorylation sites (Supplementary Fig. 2a). Phosphorylation was upregulated at 4181 sites and downregulated at 4173 sites in KRAS-mutant PDOs compared to wild-type PDOs. Ingenuity pathway analysis (IPA) identified ERK/MAPK signaling as the most significantly associated gene set (*p* = 2.51E-11; Fig. 2c). Additionally, multiple RAS-mediated signals were highly enriched, including molecular mechanisms of cancer (*p* = 6.73E-11), AMPK signaling (*p* = 2.34E-09), and integrin signaling (*p* = 2.72E-09), demonstrating the validity of the analysis. Kyoto Encyclopedia of Genes and Genomes analysis of the differentially phosphorylated proteins identified terms related to cellular proliferation, including cell cycle, base excision repair, and DNA replication (Supplementary Fig. 2b). Additionally, processes associated with cell adhesion, including tight and adherens junctions, were positively enriched in the

upregulated proteins, confirming the results of previous proteomic analyses of isogenic CRC cell lines, SW48, with heterozygous knock-in of G12D and G13D mutations[38,39]. As the role of cell adhesion in the maintenance of the ground state of ESC[40], these observations highlight the conserved role of RAS signaling in embryogenesis and CRC homeostasis.

Next, we analyzed the phosphorylation status of the KRAS-mutant PDO at 6, 24, and 72 h after trametinib treatment (Fig. 2d). K-means clustering divided the phosphorylation sites into four groups representing time-dependent alterations (Supplementary Fig. 2c). Phosphorylation sites in cluster 1 decreased over time and were enriched in the cell cycle and DNA replication (Supplementary Fig. 2d). Conversely, phosphorylation in clusters 2–4 increased over time and was enriched in the MAPK signaling pathway and processes pertaining to cell adhesion. Corresponding gene analysis has shown that signaling molecules and junctional proteins involved in the stemness of ESC are controlled by trametinib treatment of CRC PDOs[41]. Terms identified in

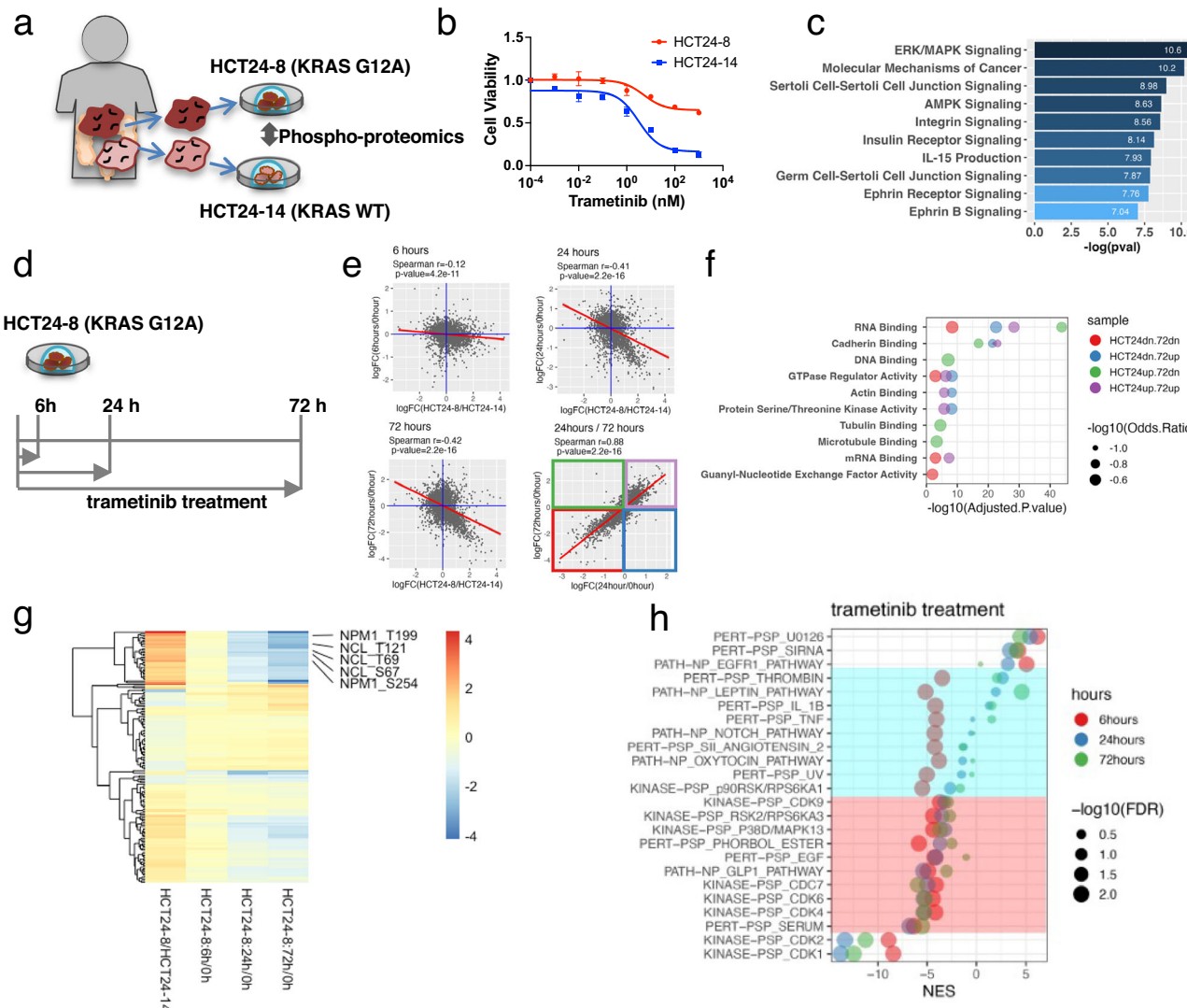

**Fig. 2 | Global phospho-proteomic analysis of PDOs. a** Overview of the comparative analysis of KRAS G12A mutant and KRAS wild-type PDOs. **b** Response of HCT24-8 and HCT24-14 to trametinib treatment. (Means ± SD, $n = 4$ wells) Source data are provided as a Source data file. **c** Gene-based enrichment analysis using datasets in Ingenuity Pathway Analysis. The bar length represents $-\log_{10}(p)$ values (one-sided Fisher's exact test). **d** Overview of trametinib treatment of PDOs. **e** Scatter plots depicting the phosphorylation sites significantly altered ($q$-value < 0.05, Welch's t-test) by KRAS mutations (x axis) and trametinib treatment (y axis). In the lower right panel, molecules differently phosphorylated at 24 and 72 h are plotted, and the frame colors corresponded to the dot colors in Fig. 2f. Correlations were assessed using Pearson's correlation coefficient with a two-sided test. **f** Gene Ontology enrichment analysis of differentially phosphorylated proteins within the GO molecular function 2023 gene sets. Genes upregulated or downregulated by KRAS mutations are indicated as HCT24up and HCT24dn, respectively, whereas those upregulated or downregulated by trametinib treatment for 72 h are indicated as 72up and 72dn, respectively. Enrichment P values were calculated using a right-

tailed Fisher's exact test, and adjusted using the Benjamini–Hochberg method. The odds ratio (OR), defined as the ratio of observed to expected gene counts within each term, reflects the magnitude of enrichment. **g** Heatmap showing the altered phosphorylation of gene sets in Gene Ontology analysis of RNA binding. Differential phosphorylation induced by KRAS mutations (HCT24-8/HCT24-14) and trametinib treatment (HCT24-8:0 h, 24 h, 72 h) is shown. Genes involved in ribosome biogenesis are annotated. Color scale on the heatmaps depicts the fold change in phosphorylation. Odds ratio and adjusted $p$-value in enrichment analysis are shown as dot plots. **h** Dot plot depicting the normalized enrichment score (NES) calculated using the single sample gene set enrichment analysis (ssGSEA) with the post-translational modifications signature database (PTMsigDB). Significantly enriched signatures (false discovery rate [FDR] <0.01) are shown. Blue mask indicates the signatures whose phosphorylation was transiently negatively enriched at 6 h after trametinib treatment, whereas the red mask indicates the signatures whose phosphorylation was constitutively negatively enriched for 72 h.

trametinib treatment were mostly shared with those identified in the KRAS mutation status (Supplementary Fig. 2b), providing evidence that trametinib counteracted the effects of KRAS-activating mutations.

Comparative analysis of trametinib treatment and KRAS mutation demonstrated that the phosphorylation sites altered by trametinib treatment were inversely correlated with those altered by KRAS mutation in a time-dependent manner (Fig. 2e). The most significant correlation was observed 72 h after treatment (Spearman's coefficient = −0.42, $p = 2.2$E-16), supporting the notion that the altered phosphorylation induced by trametinib treatment was mostly caused

by the perturbation of KRAS signaling. The proteins that were differentially phosphorylated at 24 and 72 h were highly correlated (Spearman's coefficient = 0.88, $p$-value = 2.2E-16). Interestingly, gene ontology analysis of molecular function identified RNA binding as the most enriched term, regardless of the up- or downregulation of phosphorylation (Fig. 2f). Furthermore, two molecules involved in ribosome biogenesis, Nnucleophosmin 1 and Nucleolin, were highly phosphorylated in the *KRAS*-mutant and remained unphosphorylated after trametinib treatment (Fig. 2g). The phosphorylation of these ribosome binding proteins drives liquid-liquid phase separation and

controls rDNA transcription and rRNA maturation[42,43]. These observations suggest that phosphorylation signaling is involved in trametinib-regulated ribosome biogenesis.

## Trametinib alters the phosphorylation of cyclin-dependent kinase (CDK) substrates

Next, we performed an enrichment analysis of the post-translational modification signature database (PTMsigDB) to depict site-centric molecular signatures (Fig. 2h)[44]. The strongest positive enrichment was detected for the perturbation signature of the MEK inhibitor PERT-PSP_U0126. A high enrichment score was observed at 6 h (false discovery rate [FDR] <0.001, NES = 6.16) and persisted for 72 h (FDR < 0.001, NES = 4.44), demonstrating the validity of this analysis. Conversely, two kinase-substrate signatures, KINASE-PSP-CDK1 and −2, were negatively enriched after trametinib treatment (FDR < 0.01, NES = −8.4). This result demonstrated a reduced proliferation rate in trametinib-treated cells, as these phosphor-signaling are hallmark of cell cycle progression[44]. The phosphorylation levels of RB1 at Ser807/Ser811 were reduced in the enrichment analysis, which was further confirmed by immunoblot (Supplementary Fig. 2e, f). Overall, these results demonstrated the validity of our phosphoproteomic analysis of trametinib-treated CRC organoids.

Two mTOR-related pathways, including p90RSK/RPS6KA1 signature and RSK2/RPS6KA3 signature, were identified to be negatively enriched pathways (FDR < 0.01, NES = −5.5 and FDR < 0.01, NES = −4.4, respectively). We focused on RPS6, because its phosphorylation is commonly used as readout of the mTOR signaling and found that phosphorylation levels at Ser235 and Ser240 were reduced in a time-dependent manner (Supplementary Fig. 2e). The reduced phosphorylation levels of two molecules linked to mTOR, including HSF1[45,46] and STAT1[47,48] were also reduced. These alterations of phosphorylation were confirmed by immunoblot (Supplementary Fig. 2f). Given that mTOR plays central roles in protein synthesis, these observations further validated the phosphoproteomics analysis to explore the response of CRC organoids to trametinib.

The signature sets of PTMsigDB were divided into two categories based on their kinetics. The first group exhibited transient negative enrichment at 6 h, including the perturbation signatures of IL-1B (FDR < 0.01, NES = −4.2) and TNF (FDR < 0.01, NES = −4.1) and the signature of the molecular signaling pathways of LEPTIN (FDR < 0.01, NES = −5.2) and NOTCH (FDR < 0.01, NES = −4.2) (Fig. 2h, outline in blue). The other group showed sustained enrichment over time (Fig. 2h, outline in red). Interestingly, this group included multiple kinase substrate signatures of CDKs, including CDK4, 6, 7 and 9. These results shed light on the ability of trametinib to regulate the substrate sites of other kinase inhibitors.

## Dinaciclib exerts synergistic effects with trametinib

As trametinib affects the substrate phosphorylation of multiple CDKs, we elucidated its biological significance. We hypothesized that the trametinib-induced phosphorylation of CDKs substrates is involved in the resistance of KRAS-mutated CRC. To test this hypothesis, we treated HCT24-8 cells with various CDK inhibitors, and their combination effect was evaluated by the median-effect principle[49] and its extension, the combination index equation[50]. We tested seven CDK inhibitors, and their median-effect plots for both single agents and their combination with trametinib indicated that the dose-effect data conformed to the median-effect principle of the mass-action law (Supplementary Fig. 3a). Among them, dinaciclib synergized most effectively with trametinib in reducing the viability of HCT24-8 cells (combination index <1)[50](Fig. 3a). This result was further confirmed by a nonconstant ratio combination experiment (Fig. 3b). Dinaciclib is a CDK inhibitor that targets transcription-related CDKs[51]. Transcriptome analysis revealed that the DEGs of dinaciclib-treated organoids were highly enriched in multiple

ribosome-related gene sets in GOCC, and large and cytoplasmic ribosome subunits were the most significantly enriched (p-value = 3.31E-14; Fig. 3c). The volcano plot indicated that the expression levels of the ribosomal genes were mostly decreased by dinaciclib treatment (Fig. 3d). This reduced expression of ribosome-related genes was further confirmed by IPA Canonical pathway analysis, in which eukaryotic initiation factor 2 (EIF2) signaling was identified as the most significantly enriched pathway (p.val = 6.3E-26; Supplementary Fig. 3c, d). These results suggested that dinaciclib preferentially perturbed the transcription of ribosome-related genes. Ribosomal protein genes contain a polypyrimidine initiator called the TCT motif, encompassing the transcription start sites, which confers a specialized transcription system for these genes[52]. Furthermore, RNA pol II preferentially regulates ribosomal protein expression[53]. These observations support the hypothesis that dinaciclib preferentially perturbs ribosome-related gene expression by inhibiting transcription-related CDKs. Notably, GOCC terms that were negatively enriched in dinaciclib-treated PDOs were positively enriched following trametinib treatment (Fig. 1c). This inverse correlation highlights ribosomal biogenesis as a key process related to trametinib tolerance in CRC PDOs.

## RNA polymerase I inhibitor, CX-5461, synergizes with trametinib

In addition to transcription-related CDKs, dinaciclib perturbs cell cycle-associated CDKs, which may contribute to the synergistic effects of trametinib. To clarify the role of ribosome biogenesis in the synergistic effect of trametinib, we targeted RNA pol I because its critical role in rRNA transcription is a rate-limiting step in ribosome biogenesis[54]. We used CX-5461, which is widely accepted as the first in-human inhibitor of RNA pol I[55]. Indeed, CX-5461 significantly suppressed production of nascent rRNA as detected by 5-ethynyl uridine (EU) incorporation (Fig. 3i, j) and reduced the amount of ribosome RNA rRNA (Supplementary Fig. 3e, f) expression, confirming its effects on CRC PDOs. Dinaciclib also reduced the rRNA levels, which is consistent with previous reports that rRNA transcription is coupled with ribosomal protein gene expression[56,57].

The median-effect plot demonstrated the conformity of CX-5461, both as a single agent and in combination with trametinib, to the median-effect principle of the mass-action law (Supplementary Fig. 3b). CX-5461 exhibited a synergistic effect on PDOs with trametinib (Fig. 3e), and its effects were more potent than those of dinaciclib over a wide range of concentrations (Fig. 3f). Interestingly, CX-5461 treatment strongly inhibited the expression of ribosomal protein genes (Fig. 3g, h) and EIF2 signaling gene sets (Supplementary Fig. 3g, h). Reduction in the transcription of ribosome-related genes by CX-5461 was expected, as RNA pol II-mediated transcription of ribosomal protein genes is tightly coordinated with the RNA pol I-dependent production of rRNA[24,58,59]. Collectively, these observations indicate that CRC characterized by a high ribosome biogenesis induced by trametinib are vulnerable to the inhibition of ribosome biogenesis.

Next, anti-tumor activities of CX-5461 and trametinib were evaluated in a mouse model of intestinal tumors. Autochthonous tumors were induced via the subcutaneous injection of 4-hydroxytamoxifen (4-OHT) into Apc[flox/flox], Kras[LSL-G12D/+], Lgr5-Cre[ERT2/+] mice (Fig. 4a). Trametinib significantly elevated the fibrillarin levels, recapitulating the in vitro response of PDOs to trametinib (Fig. 4b, c). No significant changes were observed in normal intestines after trametinib treatment (Fig. 4d, e).

Trametinib, CX-5461, and their combination were administrated at 13 days after 4-OHT injection once a day for 5 days, and tumor was evaluated by immunostaining of β-catenin (Fig. 4f). In control mice, 19.1% of area was identified as tumor region (Supplementary Fig. 4a, b). Trametinib and CX-5461 reduced tumor area by 8.4% and 8.2%, respectively (Fig. 4g, h). Further reduction was achieved with their combination, and the tumor was reduced to 1.3%. CX-5461 did not

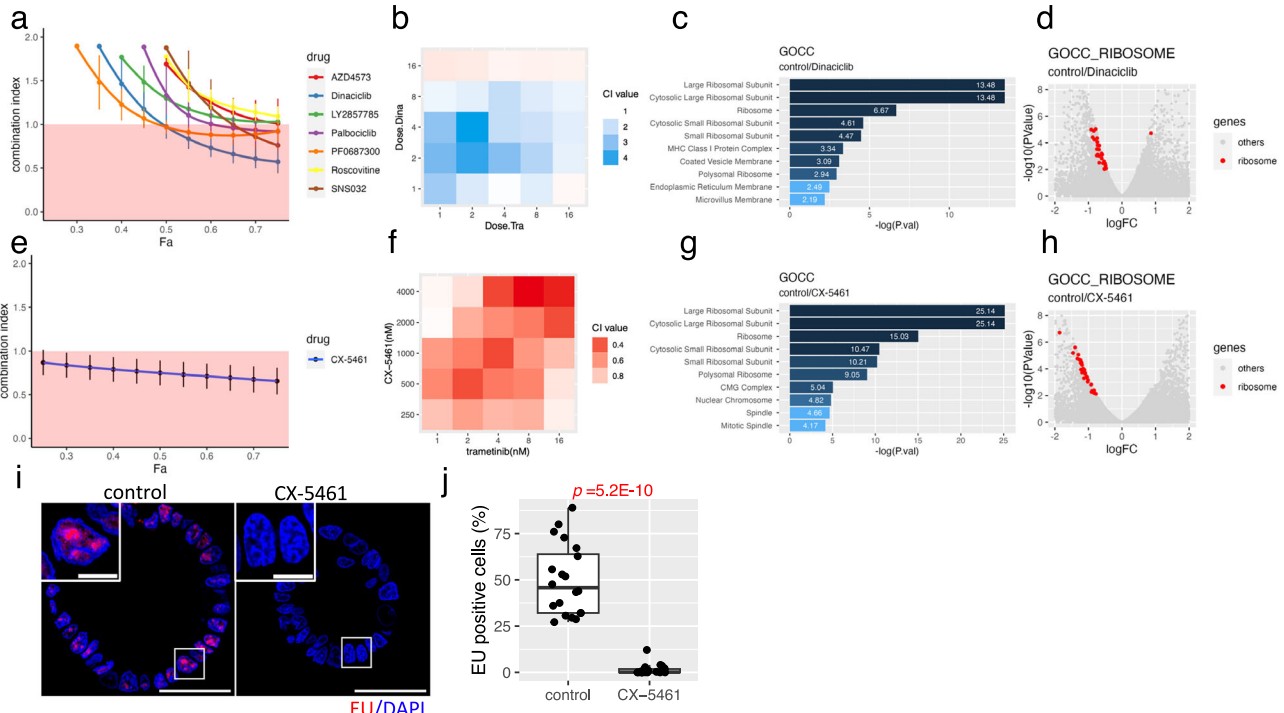

**Fig. 3 | Inhibition of ribosome biogenesis with trametinib to reduce the viability of KRAS-mutant colorectal cancer (CRC) PDOs.** Chou–Talalay plot depicting the synergistic effects of cyclin-dependent kinase (CDK) inhibitors (**a**) and CX-5461 (**e**) with trametinib. Combination index (CI) value was determined using constant-ratio drug combinations. (Means ± SD, *N* = 4 wells) Heatmap showing the synergistic effects of dinaciclib (**b**) and CX-5461 (**f**) with trametinib. CI value was determined using nonconstant-ratio drug combinations. Top 10 most significantly enriched signatures in the Gene Ontology cellular component (GOCC) in dinaciclib (**c**) or CX-5461 (**g**)-treated PDOs. Enrichment scores were calculated using differentially expressed genes (DEGs) obtained via comparison between treated and untreated PDOs using one-sided Fisher exact tests. Volcano plot showing the gene expression levels in dinaciclib (**d**)- and CX-5461 (**h**)-treated PDOs compared with those in untreated PDOs. Genes in the GOCC ribosome gene sets are shown by red

dots. Differential expression was assessed using edgeR's exact test for negative binomially distributed counts. **i** Reduced EU incorporation in CX-5461-treated CRC organoids. Representative images of EU (red) and nuclei (blue) in control and CX-5461-treated organoids are shown. High magnification views of the white boxed regions are shown in the insets. Bar = 50 μm, Bar in inset = 10 μm. **j** The percentage of EU-positive nuclei per organoid is shown in the boxplot. The central line of each box indicates the median, the box bounds represent the interquartile range (IQR), and whiskers extend to 1.5 × IQR. CX-5461 treatment markedly reduced the proportion of EU-positive cells, indicating suppression of RNA synthesis. Data of 20 PDOs obtained from four independent experiments are shown. Statistical significance was assessed using a two-sided Welch's t-test. Source data are provided as a Source data file.

---

affect EdU incorporation, but treatment with trametinib and the combination of CX-5461 and trametinib resulted in a decrease in EdU incorporation by 2.7% and 2.3% respectively (Fig. 4I, j). No significant changes in size or in EdU incorporation were observed in the normal intestines of WT mice (Fig. 4k–n) of mutant mice (Supplementary Fig. 4c–f). Notably, the combination of CX-5461 and trametinib significantly suppressed weight loss in mouse model of intestinal tumor, while having no significant effect on the body weight of wild-type mice (Fig. 4o).

In this analysis, experiments were conducted using 0.1 mg/kg of CX-5461, which is a lower concentration compared to the previously reported 35–50 mg/kg used to assess the efficacy of monotherapy[60,61]. The human equivalent dose (HED) of 0.28 mg/m²[62]. In a dose-escalation study of CX-5461 in phase I studies, patients with hematologic malignancies and solid tumors were treated at doses ranging from 25 to 450 mg/m² and 50 to 650 mg/m², respectively[55,63]. Given that the HED of CX-5461 used in combination with trametinib was markedly lower than the doses used in human clinical studies, these findings provide in vivo evidence that CX-5461 synergistically suppresses tumor growth when combined with trametinib.

### Intrinsic ribosome biogenesis is correlated with AMG510 sensitivity

AMG510 is a small compound that specifically inhibits the KRAS G12C protein and shows significant anti-tumor effects in non-small cell lung

cancer [64]. However, the objective response rate for patients with CRC was limited, and its modest efficacy warrants further evaluation as a treatment option for combination therapy[10,64]. To explore the role of ribosomal biogenesis in response to AMG510 in CRC, we established four PDOs from adenocarcinomas harboring the KRAS G12C mutation (Supplementary Table 3). Exome sequencing confirmed the KRAS G12C mutation and various other mutations frequently observed in CRC (Supplementary Fig. 5). We found that AMG510 efficiently reduced the viability of HCT191-1T, but had only a modest effect on other PDOs (Fig. 5a). UMAP visualization of single-cell expression was used to divide the cells into five clusters, and HCT191-1T contained a higher abundance of cells in C2 than other PDOs (Fig. 5b, c). Interestingly, the enrichment profile of C2 resembled that of the trametinib-induced cluster in HCT24-8 (Fig. 5d); it was highly enriched in multiple ribosome-related gene sets in GOCC and Myc targets V1 in the Hallmark data sets. The heatmap of genes listed in the cytosolic ribosome of GOCC revealed most of them are highly expressed in HCT191-1T (Fig. 5e). The gene score calculated using the cytoplasmic ribosome gene set in GOCC showed a high value for C2 (*p* = 4.3E-153) (Fig. 5f). The gene score for C5 was also significantly high (*p* = 4.9E-9). Considering that HCT191-1T had a high proportion of both C2 and C5 (Fig. 5c), these results supported the idea that the ribosome biogenesis of this organoid was elevated.

The expression profile of C2 was significantly enriched with the gene signature of naïve ground-state ESC (NES = 2.76, *p*adj = 0.01; Fig. 5g)[28]. Furthermore, it was positively enriched in the pre-

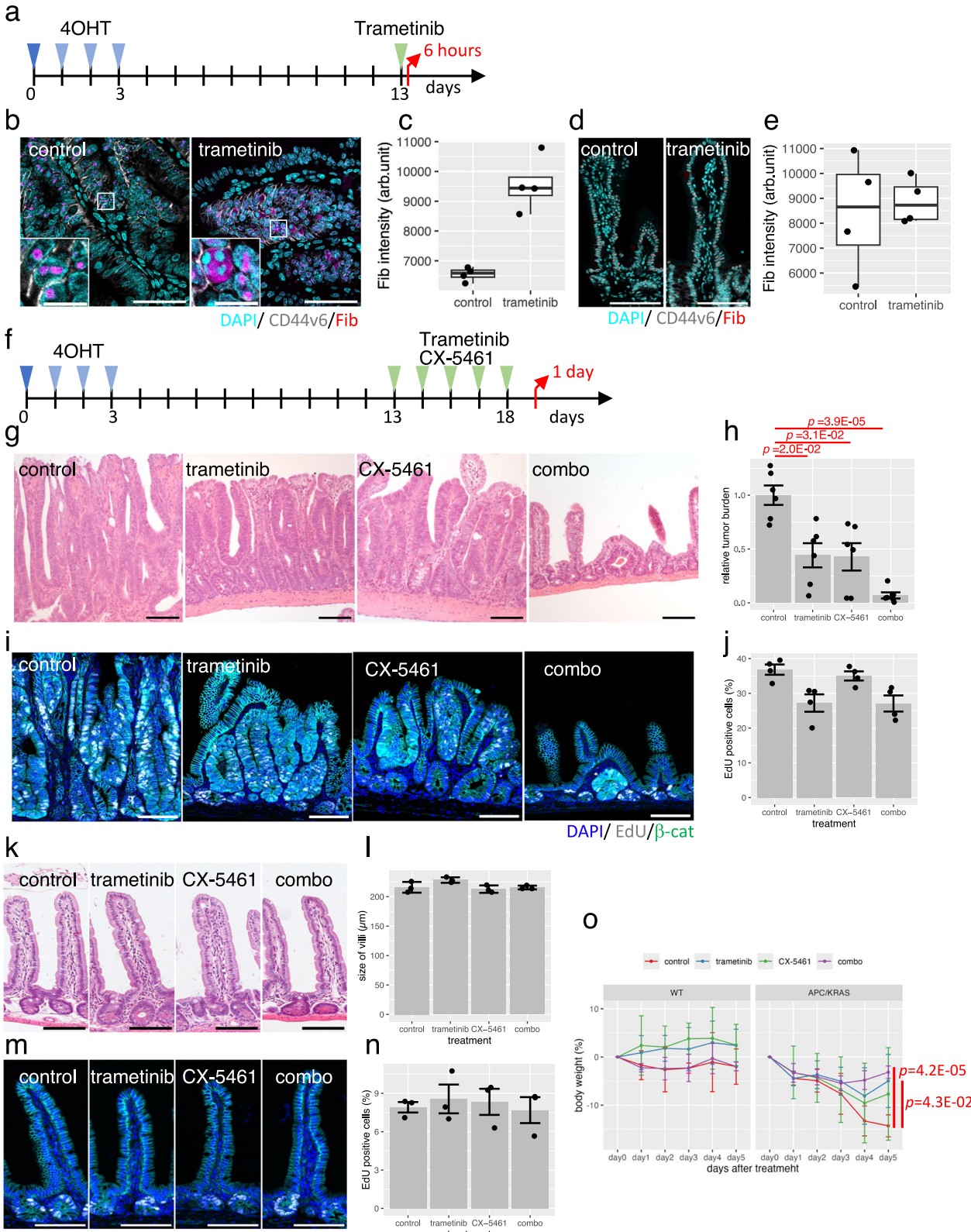

implanted ground state epiblasts (NES = 1.25, $p$adj = 0.66) but negatively enriched in the implanted primed state epiblasts (NES = −1.09, $p$adj = 0.09; Fig. 5h)[29]. Overall, PDOs sensitive to AMG510 contained a high fraction of cells with an expression profile resembling that of ground-state ESCs and epiblast.

To further evaluate differences in ribosome biogenesis among CRC organoids, we analyzed the transcription of ribosome DNA. EU

incorporation assay was employed to quantify newly synthesized RNA. Most of the EU label was detected in nucleolar, indicating that it was primarily incorporated into nascent ribosomal RNA (Fig. 5i). 3D imaging of organoids, followed by quantification of EU intensity in individual cells, revealed that HCT191-1T exhibited significantly higher incorporation compared to other organoids (Fig. 5j). Immuno-fluorescence staining of fibrillarin demonstrated that its localization in

**Fig. 4 | Evaluation of the anti-tumor effects of trametinib and CX-5461 on ApcS/S, KrasLSL-G12D/+, Lgr5-CreERT2 mice. a** Experimental design for analyzing fibrillarin expression. Female mice were used. Fibrillarin expression in intestinal tumors (**b, c**) and normal intestines (**d, e**). Representative images in tumors from vehicle (control) and trametinib-treated mice are shown (**b, d**). Bar = 50 μm, Bar in insets = 10 μm. The intensity of fibrillarin signals in each nucleus per tumors is shown in the boxplots (**c, e**). The central line of each box indicates the median, the box bounds represent the interquartile range (IQR), and whiskers extend to 1.5 × IQR. Statistical significance was assessed using a two-sided Welch's t-test ($N = 4$). **f** Experimental design for evaluation of anti-tumor effects. Experimental design for evaluation of anti-tumor effects. Analyses were performed using 4 male and 2 female mice for the trametinib treatment, and 2 male and 4 female mice for all other conditions. **g, h** Reduced tumor burden in trametinib- and CX-5461-treated mice. Representative hematoxylin and eosin (HE) images of mouse intestinal tumors treated with the indicated drugs are shown (**g**). Bar = 100 μm. Relative tumor burden is shown in the barplot (mean ± SD, $N = 6$) (**h**). $P$ values were calculated using the Brown–Forsythe test. **i, j** EdU incorporation in trametinib-treated mouse intestinal tumors. Representative images are shown (**i**). Bar = 100 μm. The percentage of EdU-positive nuclei per tumors is shown in the barplot (mean ± SD, $N = 4$) (**j**). No significant difference was observed (Brown–Forsythe test, alpha = 0.05) (**k, l**). Evaluation of normal intestine in trametinib- and CX-5461-treated WT mice. Female mice were used. Representative HE images are shown (**k**). Bar = 100 μm. Villi size is shown in the barplot (mean ± SD, $N = 3$) (**l**). No significant difference was observed (Brown–Forsythe test, alpha = 0.05). **m, n** EdU incorporation in trametinib- and CX-5461-treated WT mice. Representative images are shown (**m**). Bar = 100 μm. The percentage of EdU-positive nuclei per tumors is shown in the barplot (mean ± SD, $N = 3$) (**n**). No significant difference was observed (Brown–Forsythe test, alpha = 0.05). **o** Mouse body weight after administration of drugs. The mean and SD are shown for three WT mice and six Apc/Kras mice. $P$ values were calculated using two-sided Welch's t-test. Source data are provided as a Source data file.

nucleoli, indicating the involvement in the modification of ribosome RNA in PDOs derived from advanced CRC (Fig. 5k). 3D imaging and quantification of individual cells revealed their higher fibrillarin intensity in HCT191-1T compared to other organoids (Fig. 5l). These observations indicated that, in addition to the elevated expression of ribosomal proteins, transcriptional levels of ribosome RNA and its modification are up-regulated. Taken together, it appears that HCT191-1T has enhanced ribosome biogenesis.

### CX-5461 synergizes with the allele-specific KRAS inhibitor, AMG510

Next, we analyzed the response of KRAS G12C mutants to AMG510. AMG510 reduced EdU incorporation after treatment, and HCT191-1T was most significantly affected, consistent with the results indicating high sensitivity (Fig. 6a and Supplementary Fig. 6a). They also reduced the global protein synthesis rate as revealed by OPP incorporation (Fig. 6b and Supplementary Fig. 6b). Fibrillarin was accumulated in nuclei upon AMG510 treatment (Fig. 6c and Supplementary Fig. 6c). These observations suggested that AMG treatment reduced the global protein synthesis and upregulated the modification.

Transcriptome analysis demonstrated that AMG510 led to the enrichment of the ribosome signature in resistant clones, whereas the sensitive clone HCT191-1T was negatively enriched (Fig. 6d). Conversely, all clones exhibited positive enrichment of the MYC TARGET V1 gene sets after AMG510 treatment. Overall, similar to the trametinib treatment, AMG510-tolerant PDOs adopted a cell state with high ribosome biogenesis, yet the global protein synthesis rate is low. Interestingly, PDOs sensitive to AMG510 had a highly intrinsic ribosome signature and did not show enhanced expression in response to AMG510 treatment. These findings highlight the role of ribosome biogenesis in the response to AMG510, although the correlation needs to be carefully evaluated using a large number of PDOs with KRAS G12C mutations.

Finally, we evaluated the synergistic effects of CX-5461. CX-5461reduced the incorporation of EU (Supplementary Fig. 6d). The CRC organoids exhibited different sensitivities to CX-5461 (Fig. 6f). Nevertheless, it exhibited a synergistic effect on AMG-510-tolerant clones (Fig. 6g). Nonconstant CI estimation confirmed their robust synergistic effects at various concentrations (Fig. 6h). Taken together, these results suggest ribosome biogenesis as a promising target for combination therapy with AMG510 for patients with CRC with the KRAS G12C mutation.

### Discussion
In this study, we found that inhibition of the KRAS pathway promoted the cell states in KRAS-mutant CRC organoids, characterized by high levels of ribosome biogenesis, yet global translation rate was low. These cellular states are well-documented in embryonic and adult stem cells[17].

MEK inhibitor reprograms primed ESCs into naïve ground state ESCs[25]. We also found KRAS inhibitor-treated PDOs were highly enriched in Myc signature, which controls ribosome biogenesis and protein synthesis to regulate the reprograming of ESCs[65–67]. Finally, the expression profile of KRAS-inhibited CRC was highly correlated with that of ground-state ESCs. As ground-state naïve pluripotency is established from the pre-implanted epiblast of the mature blastocyst, whereas primed ESCs are in vitro counterparts of post-implanted primed epiblasts[25], we believe that advanced CRC cells are equipotent enough to enter the embryo-like ground state and exploit the cellular program of early embryogenesis to become tolerant to KRAS inhibition.

Ribosome biogenesis correlates with the proliferation rate across diverse cell types and is important for the self-renewal of stem cells[17]. Consistently, Batlle et al.[35] showed that rRNA transcription and protein synthesis are elevated in a subset of tumor cells, which defines the stemness of CRC cells[18]. Protein synthesis is the most energy-consuming process in cells, and its low rate is the key to effectively maintain the stem cell pool[68–70]. During development, stem cells differentiate in response to developmental and environmental cues. To achieve this response accurately and rapidly, high ribosome biogenesis and low protein synthesis rates are needed for the removal of old proteins and their replacement with new proteome profiles suitable for the functions of differentiated cells[71–73]. Similar to these developmental processes, our analysis of PDOs reveal that CRC cells with KRAS mutation acquire cellular plasticity by altering ribosome biogenesis and protein synthesis to become tolerant to RAS inhibition. Considering that CRC patients comprise a population with diverse genetic background and somatic mutations, analyzing a greater number of PODs is expected to elucidate more detailed molecular basis of tolerance to KRAS inhibition.

Generally, tumor cells exhibit increased protein synthesis rates to facilitate the robust metabolic processes necessary for sustained growth and proliferation; therefore, ribosome biogenesis is considered a potential therapeutic target[74–77]. Transcription of rRNA is a rate-limiting step in ribosome biogenesis, and Myc plays a critical role in RNA Pol I-mediated transcription. Myc activation drives malignancy and contributes to the hyperactivation of rDNA transcription. CX-5461 is a first-in-class therapeutic agent that targets RNA pol I inhibitors and shows efficacy against Myc-driven tumors[78–82]. It also efficiently suppresses tumor growth in a mouse model of intestinal tumors[83]. A phase I dose-escalation study in advanced hematological cancers revealed that CX-5461 is safe at doses associated with clinical benefits[55]. These studies provide a rationale for using CX-5461 as a therapeutic agent along with KRAS inhibitors. However, CX-5461 stabilizes G4-quadruplex and impedes topoisomerase II activity, resulting in synthetic lethality in *BRCA1*- and *BRCA2*-mutated cancers[84–86]. Moreover, CX-5461 causes collateral mutagenesis[87]. These reports should be carefully considered for future clinical applications.

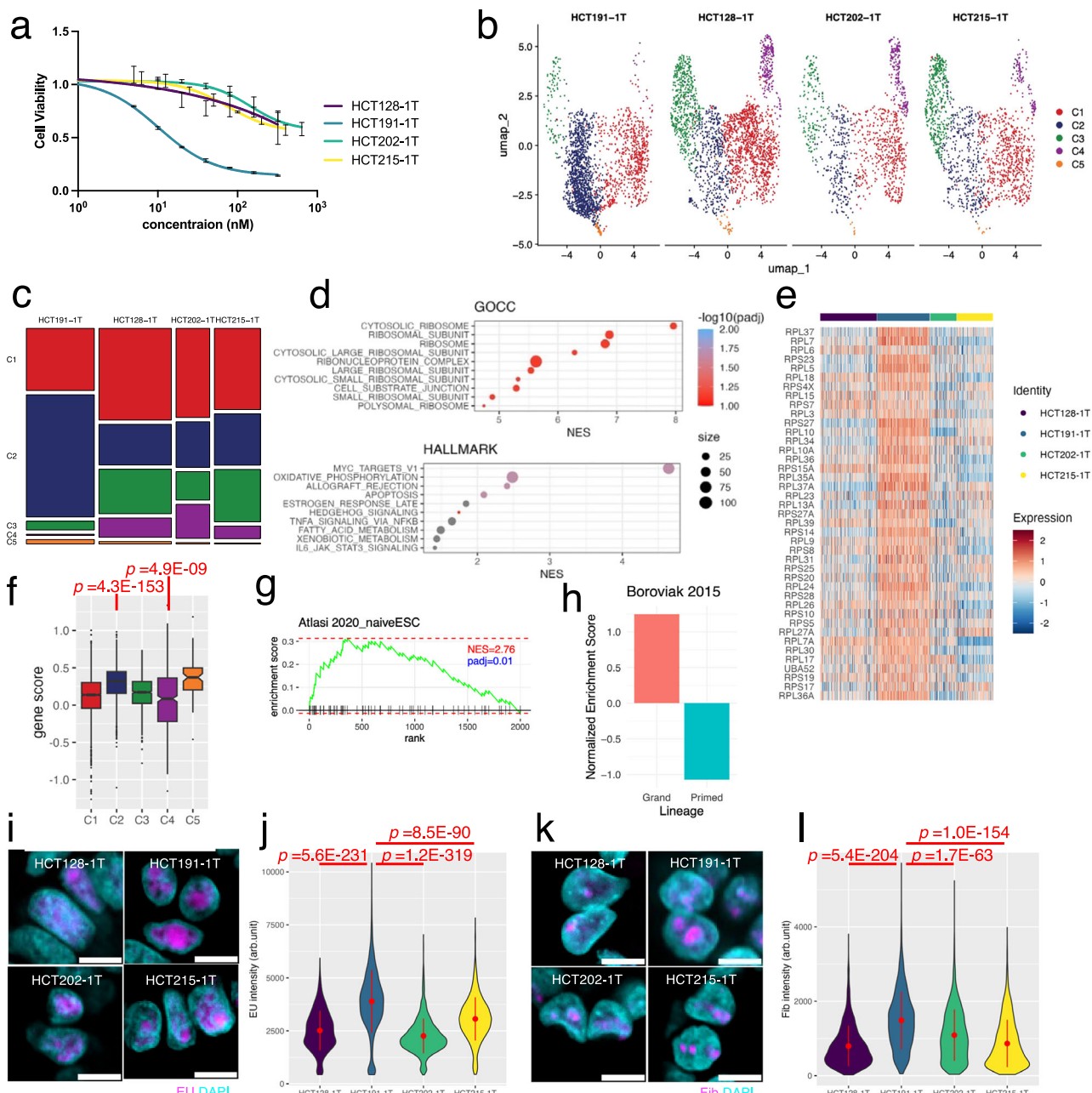

**Fig. 5 | Correlation between AMG510 sensitivity with ribosome biogenesis.**
**a** Response of KRAS G12C organoids to AMG510. Cell viabilities of organoids were evaluated at 72 h after treatment with indicated concentrations of AMG510 (mean ± SD, $N = 4$ wells). **b** UMAP of PDOs. The indicated PDOs were labeled with cell multiplexing oligos and subjected to single-cell RNA sequencing (scRNA-seq). Cells colored by cluster are shown. A total of 2143 cells from HCT191-1T, 2285 cells from HCT128-1T, 1062 cells from HCT202-1T, and 1451 cells from HCT215-1T were analyzed. **c** Proportion of cells in each cluster of CRC organoids. **d** Enrichment analysis of C2 with GOCC (upper panes) and Hallmark (lower panel). Gene set enrichment was assessed as described in Fig. 1c. Top 10 most significantly enriched pathways are shown in the order of NES. Heatmap (**e**) and boxplot (**f**) depicting the different expression levels of genes listed in cytosolic ribosome of GOCC among CRC organoids. The central line of each box indicates the median, the box bounds represent the interquartile range (IQR), and whiskers extend to 1.5× IQR. Statistical comparisons were performed using a two-sided Welch t-test. Gene set enrichment analysis of C2 with ground-state ESCs (**g**)[28] and epiblast (**h**)[29]. Gene set enrichment was assessed as described in Fig. 1c. **i**, **j** Different EU incorporation rate among CRC organoids. Representative images of the EU (magenta) and nuclei (cyan) of the indicated organoids are shown (**i**). Bar = 10 μm. The intensity of EU signals in each cell is shown in violin plot (**j**). Data of 2000 cells obtained from four independent experiments are shown. The central red dot and error bars indicate the mean ± SD. Statistical comparisons were performed using a two-sided Welch's t-test.
**k**, **l** Accumulation of fibrillarin in the nucleolus. Representative images of fibrillarin (magenta) and nuclei (cyan) of the indicated organoids are shown (**k**). Bar = 10 μm. The intensity of fibrillarin signals in each cell is shown in violin plot (**l**). Data of 2000 cells obtained from four independent experiments are shown. The central red dot and error bars indicate the mean ± SD. Statistical comparisons were performed using a two-sided Welch's t-test. Source data are provided as a Source data file.

Here, we established four PDOs with the KRAS G12C mutation in patients with advanced CRC. As anticipated from the modest objective response of CRC patients in a clinical study[64], three clones were resistant to AMG510. These clones showed increased ribosome biogenesis with AMG510 and exhibited significant synergistic effects with CX-5461. In contrast to the resistant clones, one sensitive clone contained a high proportion of cell clusters with intrinsically high ribosome biogenesis, and AMG510 did not increase the ribosome

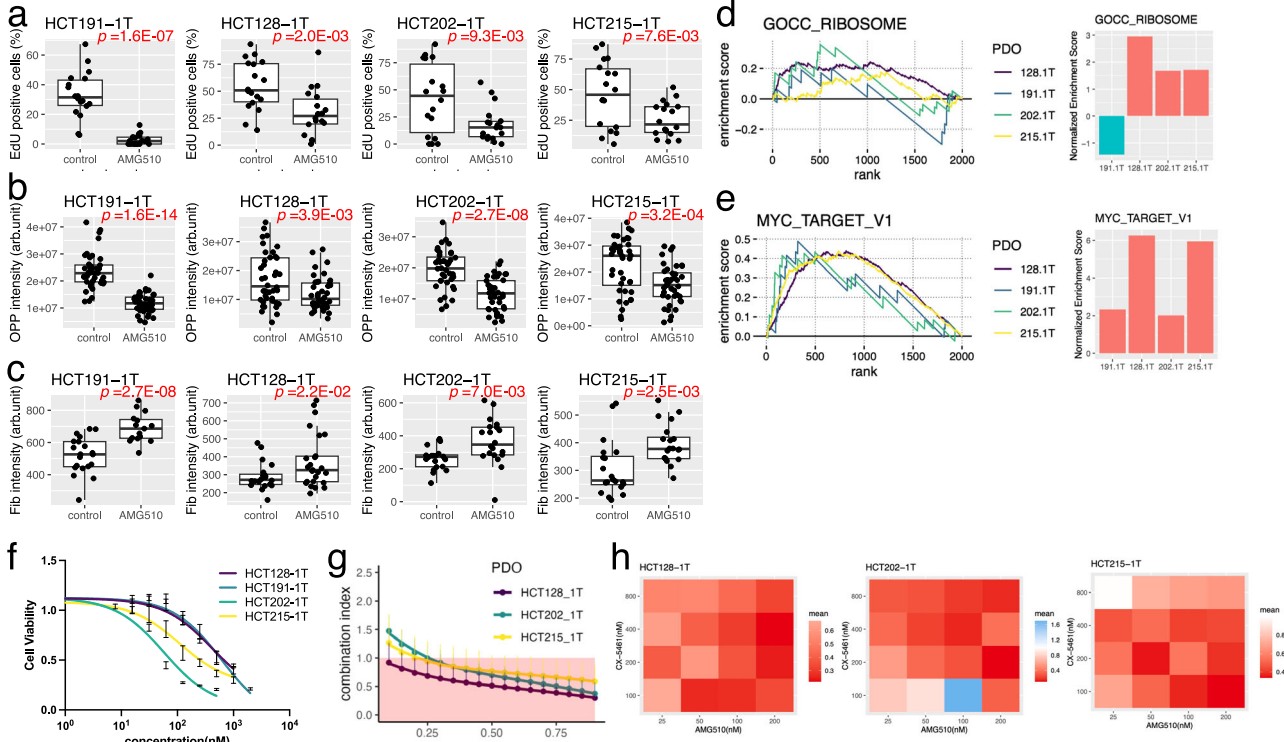

**Fig. 6 | Synergistic effects of CX-5461 and AMG510 in reducing the viability of KRAS G12C mutant PDOs.** Effects of AMG510 treatment on EdU incorporation (**a**), global protein synthesis (**b**), and fibrillarin expression (**c**) in KRAS G12C CRC organoids. The percentage of EdU-positive nuclei per organoid (**a**), the average intensity of OPP signals per cell in each organoid (**b**), and fibrillarin expression per organoid (**c**) are shown in the boxplots. The central line of each box represents the median, the box boundaries indicate the interquartile range (IQR), and the whiskers extend to 1.5 × IQR. Data from 18 organoids (**a**) or 20 organoids (**b**, **c**), obtained from four independent experiments, are shown. *P* values were calculated using a two-sided Welch's two-sample t-test. GSEA of AMG510-treated and untreated organoids using GOCC RIBOSOME gene sets (**d**) and MYC TARGET V1 signature (**e**).

GSEA plots (right panels) and bar plots indicating the NES (right panels) are shown. **f** Response of KRAS G12C organoids to CX-5461. Cell viabilities of organoids were evaluated at 72 h after treatment with indicated concentrations of CX-5461. Mean ± SD from four independent experiments is shown. **g** Chou–Talalay plot depicting the synergistic effects of CX-5461 with AMG510 in PDOs exhibiting resistance to AMG510 monotherapy. CI value was determined using constant-ratio drug combinations. (Means ± SD, *N* = 3 biological replicate) **h** Heatmap showing the synergistic effects of CX-5461 and AMG510 in PDOs exhibiting resistance to AMG510 monotherapy. CI value was determined using nonconstant-ratio drug combinations. Source data are provided as a Source data file.

signature. If this clone represents patients who respond to AMG510 treatment, our observations suggest that the intrinsic level of ribosomal biogenesis provides clues for the development of molecular markers to predict the efficacy of AMG510. However, PDOs should be established from a large number of patients with the KRAS G12C mutation, and the correlation between intrinsic ribosome biogenesis and sensitivity to AMG510 should be analyzed to verify our findings.

Different KRAS mutations exhibit varying strength of phenotype and responsiveness against chemotherapeutic agents. This is exemplified by the KRAS G13D mutation, which has been clinically proven to be sensitive to cetuximab[88]. Two proteomic studies reported mutation-specific phosphorylation signature between KRAS G12D and G13D[38,39]. Considering the recent development of mutation-specific and pan-mutant KRAS inhibitors, clarifying the differences among KRAS mutations is of particular interest. Establishing PDOs with different KRAS mutations and conducting comparative analyses will be of highly important experiments in the future.

*KRAS* is the most commonly mutated oncogene, and development of allele-specific inhibitors is a promising strategy for cancer treatment[89]. AMG510 was the first drug targeting the *KRAS*-mutant proteins that was approved for clinical use[64]. However, it exhibits modest single-agent anti-tumor activity in CRC; a multi-arm clinical study to evaluate the combination of AMG510 with other therapies is currently underway[10]. Our findings suggest that targeting ribosome biogenesis in combination with AMG510 may represent a potential

therapeutic strategy in PDOs. KRAS G12C mutation occurs in approximately 4% of all CRC cases[90], but therapeutic agents targeting KRAS G12D[89] or pan KRAS mutations[5] are currently being tested. More recently, small compounds targeting GTP-bound active state of KRAS proteins are developed[91,92]. As KRAS mutations are observed in 40% of all CRC cases[90], our findings can aid in the development of novel treatment options for many patients with CRC.

## Methods
### Establishment of colorectal tumor organoids
HCT24-8 and -14 PDOs were established and characterized in a previous study[21]. To establish PDOs harboring the KRAS G12C mutation, CRC specimens were collected via surgical resection and histological diagnosis was performed by an experienced pathologist. PDOs were established as previously reported[93]. Briefly, surgical specimens were washed with phosphate-buffered saline (PBS) (Thermo Fisher Scientific) and minced into 1 mm³ fragments using surgical scissors. The fragments were digested with a digestion buffer (Dulbecco's modified Eagle's medium [DMEM], fetal bovine serum, penicillin/streptomycin, collagenase, and dispase) (Thermo Fisher Scientific) at 37 °C for 60 min. Isolated tumor cells were embedded in Matrigel droplets and overlaid on a culture medium, as previously described[93]. Advanced DMEM/F12 was supplemented with penicillin/streptomycin, 10 mM HEPES, 2 mM GlutaMAX, 1× B27 (Thermo Fisher Scientific), 10 nM gastrin I (Sigma), and 1 mM N-acetylcysteine (Wako) to prepare the basal culture medium. The ENR (EGF, Noggin, and R-spondin1)

medium was prepared by supplementing the basal culture medium with the following niche factors: 50 ng/mL of mouse recombinant EGF (Thermo Fisher Scientific), 10% Noggin conditional medium[94], and 1 mg/mL of human recombinant R-spondin1 (R&D). The medium was changed every 3–4 days.

### Exome sequencing

Genomic DNA was extracted using the QIAamp DNA Mini Kit (Qiagen), and its quality was validated using the Qubit system (Thermo Fisher Scientific). Exome sequencing was performed by Macrogen Japan Corp. (Tokyo, Japan). Genomic DNA was fragmented and used to prepare 151 bp paired-end libraries using the SureSelect Human All Exon V6 kit (Agilent). Sequencing was performed using the Illumina HiSeq2500 system. Fastq files were analyzed using the Genomon pipeline provided by the Human Genome Center, Institute of Medical Science, University of Tokyo.

### PDO viability assay

Trametinib and AMG510 were purchased from Sellec (Kanagawa, Japan). Organoids were digested with TrypLE Express (Thermo Fisher Scientific) supplemented with 10 μmol/L Y27632 at 37 °C for 15 min with pipetting every 5 min. The cells were suspended in a basal medium, and clumps were removed by passing the suspension through a 40-μm cell strainer. The suspension was centrifuged at $1000 \times g$ for 5 min, and the cells were resuspended in the basal medium. Cells were counted and adjusted to $2.0 \times 10^5$ cells/mL in the Matrigel. Then, 10 mL of cell suspension were dispensed into each well of U-bottom 96-well microplates (Thermo Fisher Scientific). PDOs were cultured for three days to allow the formation of organoids and then treated with drugs at the indicated doses for 7 days. The medium was changed every 3–4 days. We generated 8-step, ten-fold drug matrices in technical quadruples. Cell viability was measured using a CellTiter-Glo 3D Cell Viability Assay Kit (Promega). The readings were obtained using a Mithras LB 940 luminometer (Berthold Technologies). In the 96-well plate, the average cell viability at each drug concentration was calculated using DMSO-treated cells as controls. Data were analyzed using the GraphPad Prism 9 software (GraphPad).

Titrations to determine drug synergy were performed by plating the PDOs in 96-well plates as described above. After 3 days, PDOs were treated with a combination of trametinib and various CDK inhibitors, or CX-5461 (Fig. 3), or AMG510 and CX-5461 (Fig. 5). The drugs were titrated in a seven-dose manner, ranging from four times the IC50 concentration to one-sixteenth of the IC50 concentration. To determine synergy in a nonconstant ratio, drugs were titrated in a four-dose manner, ranging from twice the IC50 concentration to one-fourth the IC50 concentration. Cell viability was determined using the CellTiter-Glo 3D Cell Viability Assay, as described above. The synergistic effects were determined using the median effect principle proposed by Chou and Tallalay[95]. R package was obtained from the following website: https://rdrr.io/github/snowoflondon/CIcomputeR/.

### RNA sequencing

PDOs were treated with 100 nM AMG510 for 24 h. Total RNA was extracted using an RNeasy mini kit (Qiagen) and quantified using 2100 Expert Bioanalyzer software (Agilent). RNA-seq analysis was outsourced to Macrogen Japan Corp. (Tokyo, Japan). Following the manufacturer's instructions, total RNA with an RIN > 7 was used to generate a barcode-labeled library using an Illumina TruSeq Stranded mRNA LT Sample Prep Kit (Illumina). Sequencing of 150 bp paired-end reads was performed using the Illumina NovaSeq6000 system (Illumina).

Salmon was used for quantifying transcript abundance from RNA-seq reads[96]. Tximport package was used to import transcript-level abundances and summarizes abundances, counts, and transcript

lengths to gene-level matrix (https://github.com/thelovelab/tximport). Gene enrichment analysis was performed using the fgsea package with nperm=1000 [97] (https://bioconductor.org/packages/release/bioc/html/fgsea.html). Tidy format gene sets were retrieved using msigdbr package (https://cran.r-project.org/web/packages/msigdbr/vignettes/msigdbr-intro.html), and the rnk file was generated using wilcoxauc function in the presto package (https://rdrr.io/github/immunogenomics/presto/man/wilcoxauc.html).

**Single-cell RNA sequencing.** HCT24-8 PDOs were cultured for 3 days. PDOs harboring the KRAS G12C mutation were cultured for 3 days, followed by treatment with 10 nM trametinib for 24 h, or left untreated. To analyze KRAS G12C mutant PDOs, cells were collected 3 days after splitting. Multiplexing was performed using the 10X Genomics 3' CellPlex Kit (PN-1000261). Organoids were collected, and single cells were prepared using TrypLE (Thermo Fisher Scientific). Cells were suspended in PBS containing 0.25% BSA and centrifuged at $1000 \times g$ for 10 min. After removing the supernatant, cells were resuspended in 100 μL of Cell Multiplexing Oligo and incubated for 5 min at RT. Cells were then washed with PBS containing 0.25% BSA three times. Cells were then resuspended with 100 μL CellPlex Oligo and incubated for 5 min. Three multiplex pools were then generated. Pool1 contained trametinib-treated and untreated HCT24-8. Pool2 contained HCT191-1T and HCT-215. Pool3 included HCT128-1T and HCT201-1T. The Chromium Nex GEM Single Cell 3' Kit v.3.1 was used to encapsulate single cells. To target 20,000-cell recovery for each pool, cells were concentrated to approximately 800 cells/μl. Pooled cells were loaded into one lane of the Chromium chip, and the single-cell RNA sequencing (scRNA-seq) library was prepared using the 10X Genomics Chromium protocol. Libraries were sequenced using an Illumina NextSeq 550 platform (Illumina) with a 150-bp paired-end run.

### Analysis of scRNA-seq data

Sequence data were performed using Cell Ranger (v.6.1.1), and Seurat was used for downstream analysis[98]. The data were filtered to remove cells with fewer than 500 and over 9000 unique genes per cell, and over 20% of mitochondrial expression. The filtered data were processed with the Seurat pipeline using SCTransform, followed by principal component analysis using RunPCA. Cell clustering was conducted with the FindNeighbors function with 1:30 dimensions and the FindClusters function with a resolution of 0.2. The cells were projected onto the UMAP embedding space using RunUMAP at 1:30 dimensions. To obtain cluster-specific expressed genes, we used the FindAllMarkers function with the options of only pos = TRUE, min.pct = 0.1, and logfc. threshold = 0.1. AddModuleScore was used to calculate gene score.

To annotate clusters, gene overlap analysis between cluster-specific expressed genes and large intestine markers, developed using the single-cell transcriptomic atlas of humans[23] was performed using the newGeneOverlap function in the GeneOverlap package (https://bioconductor.org/packages/release/bioc/html/GeneOverlap.html). Gene enrichment analysis was performed as described in the RNA-seq section.

### Phospho-proteomic analysis

Cells were dissolved in PTS buffer (12 mM sodium deoxycholate, 12 mM sodium lauroyl sarcosinate, and 50 mM ammonium bicarbonate). The samples were reduced with 10 mM TCEP, alkylated with 20 mM iodoacetamide, and quenched with 21 mM of L-cysteine, following digestion with trypsin (protein weight: 1/50) and Lys-C (protein weight: 1/50) for 16 h at 37 °C. Samples were acidified with 1% TFA and centrifuged at $20,000 \times g$ for 10 min at 4 C. Supernatants were desalted, and an IMAC column was used for phosphopeptide enrichment[99]. The phosphopeptides were labeled by the TMT 11plex reagent

(Thermo Fisher Scientific) and quenched with hydroxylamine. The TMT-labeled peptides were mixed, desalted, and fractionated on C18-SCX StageTips[100]. Phosphotyrosine enrichment was performed using the pY1000 antibody as previously reported[101,102]. The time course experiment was performed in biological duplicate. Other experiments were performed in triplicate.

The nano-LC gradient was performed at 280 nL/min and consisted of a linear gradient of buffer B from 5 to 30% B over 135 min for phosphoproteome analysis and 45 min for phosphotyrosine proteome analysis. The Q Exactive or Q Exactive Plus instrument was operated in a data-dependent mode[100]. Phosphopeptide identification was conducted using MaxQuant 1.5.1.2 or 1.6.3.3, and the UniPort Human Database (01/2017 release) was supported by the Andromeda search engine. Phosphorylated and non-phosphorylated peptides were distinguished by the mass shift corresponding to the phosphorylation modification (molecular weight = 79.966) at the MS1 level. Additionally, MS2 scans (fragmented ionic products of phosphopeptides) were used to determine the localization of phosphorylation in the phosphopeptide sequence. The PIF filter was set at 75%. The search results were filtered to a maximum (FDR) of 0.01 at the protein, peptide-spectrum match, and posttranslational modification (PTM) site levels. Kinase activity prediction was performed using site-centric post-translational modification-signature enrichment analysis (PTM-SEA)[44], a seven-amino-acid sequence flanking the phosphosite as an identifier and the human kinase/pathway definitions of PTMsigDB (v.1.9.0) according to the following parameters: (gene.set.database = "ptm.sig.db.all.flanking.human.v1.9.0. gmt", sample.norm.type = "rank, "weight = 0.75, statistic = "area.under. RES", output.score.type = "NES, "nperm = 1000, global.fdr = TRUE, min.overlap = 3, correl.type = "z.score").

### Whole-mount immunofluorescence analysis of organoids

Organoids were embedded into Matrigel, and 20 µl of droplets were placed into 48-well plate. After three days of culture in ENR medium, the organoids were treated with 10 nM trametinib or 100 nM AMG510 for 24 h. The Matrigel was removed by gentle pipetting in a cell recovery solution (Corning) and replated onto an optically clear flat-bottom 96-well plate (PhenoPlate, PerkinElmer). After centrifugation at $1000 \times g$ for 5 min, organoids were fixed in 4% PFA in PBS for 30 min at room temperature and permeabilized with 0.5% Triton X-100 in PBS for 1 h. Rabbit anti-fibrillarin (1:500; ab5821; Abcam) was used as the primary antibody, and cy3 conjugated goat anti-rabbit IgG (1:1000, AP132C; Merck Millipore) was used as the secondary antibody. Cell nuclei were stained with 10 µg/mL 4′6-daamidino-2-phenylindole (DAPI; Sigma) in PBS for 15 min. For EdU and OPP staining, we used the Click-iT EdU cell proliferation kit and Click-iT Plus OPP protein synthesis kit, respectively, with Alexa Fluor 488 dye (Thermo Fisher Scientific) according to the manufacturer's instructions. For EU staining, we used the Click-iT Nascent RNA Capture kit with Alex Fluor 594 dye (Thermo Fisher Scientific) according to the manufacturer's instructions.

High-throughput imaging was performed on a Yokogawa CV8000 instrument equipped with a CSU-W1 spinning disk. Nine fields were scanned for each well with a x4 objective lens, and the images were analyzed using custom-written CellPafhfiner software (Yokogawa) to obtain individual organoid positions. Coordinated fields were imaged with x40 objective lens, and three dimension imaging were acquired with 100 µm of z-plane and 2 µm z-steps.

### Immunoblot analysis

Organoids seeded in 24-well plates were harvested after 6-h incubation with trametinib in PTS buffer supplemented with protease inhibitors (Complete Mini, Roche) and phosphatase inhibitors (PhosSTOP, Roche), followed by sonication (Branson Sonifier, Heinemann). Protein concentrations of supernatants were measured using a BCA assay kit (Thermo Fisher Scientific). Lysates were mixed with an SDS-loading buffer and heated to 100 °C for 5 min. Proteins were separated by SDS-PAGE and transferred to a nitrocellulose membrane. Membranes were blocked with 5% (w/v) skim milk in PBS. Antibodies against PABP1 (x1000, Proteintech 10970-1) and RPL13A (x1000, Proteintech 14633-1-AP), RPL23 (x1000, Proteintech 16086-1-AP), and β-actin (x1000, Sigma A1978) were used. Goat anti-rabbit HRP conjugate (x1000, BioRad 1705046) or Goat anti-mouse HRP conjugate (x1000, BioRad 1705047) were used as secondary antibody. ECL Western blotting substrate (BioRad 1705061) was used for visualization of bands.

### Animal procedures

To generate *Lgr5-CreERT2* ; *Apc*^S80S/S; *Kras*^G12D/+ double conditional knockouts mice, *Apc*^S80S/S; *Kras*^G12D/+ mice were crossed with Lgr5-CreERT2; *Apc580*^S/S mice. For genotyping, DNA was extracted from tail tips, and alleles were genotyped using standard PCR with Taq DNA polymerase (Millipore Sigma). Allele-specific primers are shown in Supplementary Table 4. All mice were 10 weeks old at the start of the experiment. Sex was not considered in the study design or analysis, as preliminary experiments showed no significant sex-dependent differences in the measured outcomes. In vivo recombination was induced by subcutaneous injection of 4-hydroxytamoxifen (4OHT, Sigma-Aldrich) at 40 mg/kg/day dissolved in corn oil, followed by three consecutive injections at 20 mg/kg/day. Mice were assigned to control group or treated with trametinib (1 mg/kg), CX-5461 (100 µg/kg) or their combination. Treatment with chemotherapeutic drugs was initiated 13 days after recombination by injection of 4OHT and was administered for 5 days until the day before the mice were killed. If the body weight loss exceeded 20% or abnormal behavior was observed, the experiment was terminated in accordance with the institutional guidelines.

For histological analysis, intestines were fixed in 10% neutral-buffered formalin (Wako) for 24–48 h and stored in 100% ethanol for 24 h. Swiss rolls were dehydrated via successive incubation with 100% ethanol for 2 h 7 times and then with xylene for 2 h 2 times. Subsequently, they were embedded in paraffin, and sections were cut at 3 µm with microtome.

For Immunofluorescence analysis, slides were deparaffinized by successive incubation in 100% xylene four times and were subsequently rehydrated by following incubation; 100% ethanol six times. For heat-induced epitope retrieval, slides were cooked in 10 mM sodium citrate (pH 6.0) in a microwave oven for 5 min, cooled gently, and rinsed with water. Slides were incubated with rabbit anti-fibrillarin (1:500; ab5821; Abcam) or rat anti- CD44v6 antibody (1:400, MCA1967; BioRad) for overnight at 4 °C. Subsequently, the slides were incubated with Cy3-conjugated goat anti-rabbit IgG (1:200, AP132C; Millipore) or Cy5-conjugated anti-rat IgG (1:200, 712-175-153; Jackson) for 1 h at room temperature. Cell nuclei were stained with 10 µg/mL DAPI in PBS for 30 min. For EdU staining, 200 µg of EdU was intraperitoneally injected 1 h before killing, and the Click-iT EdU cell proliferation kit for imaging, Alexa Fluor 647 dye (Thermo Fisher Scientific) was used to detect EdU-incorporated cells.

For histological analysis, slides were deparaffinized, rehydrated as described above, and stained with H&E for histological analysis. For immunohistochemistry, epitopes retrieval was performed as described above, and slides were incubated with mouse anti-β-catenin (1:1000, 610154, BD Biosciences) overnight at 4 °C. The primary antibodies were visualized using the Envision+ system-HRP labeled polymer (DAKO) and 3.3′-diaminobenzidine (DAKO) according to the manufacturer's instructions. To score the number and size of the adenomatous lesions, images were obtained using an AxioImager M2 microscope (Zeiss), and tiled images were created and scored using the MosaiX and AutoMeasure Modules provided with the AxioVision4 program (Zeiss).

## Statistical analyses

Drug responses and the results of the linear regression model and Pearson correlation analyses of phosphorylated proteins were analyzed using GraphPad Prism (GraphPad Software, Inc.) and R software packages. Data for each experimental PDO are expressed as the mean ± standard error of the mean. Confidence intervals of 95% or more were considered significant. For further statistical details, please refer to the figure legends.

## Ethics statement

All experiments were performed in accordance with the relevant guidelines and regulations. All the samples and methods used in this study were approved by the Ethics and Medical Research Committee of the Japanese Foundation for Cancer Research, Tokyo, Japan (approval number: 2013-1105). Clinical samples used for organoid establishment and biological analyses were obtained from patients with CRC at the Cancer Institute Hospital, Tokyo, Japan. Informed consent was obtained from all patients. The animal experiments were approved by the Animal Experimental Committee of the Japanese Foundation for Cancer Research, Tokyo, Japan (approval number: 10-03-26).

## Reporting summary

Further information on research design is available in the Nature Portfolio Reporting Summary linked to this article.

## Data availability

The raw RNA-seq data generated in this study have been deposited in the European Nucleotide Archive (ENA) database under accession code E-MTAB-16177 [sample entry: https://www.ebi.ac.uk/ena/browser/view/SAMEA120567684?show=reads] and E-MTAB-16244 [project entry: https://www.ebi.ac.uk/ena/browser/view/ERP185461]. The processed data from the scRNA-seq are available in the Gene Expression Omnibus (GEO) database under accession code GSE264485. The raw mass spectrometry (MS) data generated in this study have been deposited in the ProteomeXchange Consortium (https://www.proteomexchange.org/) via the Jpost partner repository under accession ID PXD052914/JPST003164 [https://repository.jpostdb.org/entry/JPST003164.0]. The raw exome-seq data are protected and are not available due to data privacy laws, and the processed data generated in this study are provided in the Supplementary Information file. Source data are provided with this paper. The data from this study are available from the corresponding author upon reasonable request.

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

## Acknowledgements

We are grateful to all participants of this study. We would like to thank all lab members and staff for their assistance with sample collection and completion of this study. This work was supported in part by MEXT/JSPS KAKENHI (grant numbers 21H02770 [R.Y.] and 22K19468) and AMED (grant number JP23ama221116).

## Author contributions

Y.N., H.O., E.S., and S.N. established the PDOs. Y.T., H.Y., Y.N., H.Y., and D.K. conducted the in vitro analysis of PDOs. M.S. conducted in vivo analysis. N.K., Y.A., and J.A. conducted proteomic analysis. K.K. and R.M. conducted data analysis and visualization of the transcriptome. K.O., S.M., and Y.K. supervised the project. Y.K., S.N., R.M., and R.Y. designed the study. R.Y. wrote and edited the manuscript.

## Competing interests

The authors declare no competing interests.
