## [Transparent Peer Review File · Nature Communications]

Ribosome biogenesis as a potential therapeutic target in KRAS mutant colorectal cancer.

Corresponding Author: Dr Ryoji Yao

Version 0:

Reviewer comments:

Reviewer #1

(Remarks to the Author)

This is an excellent manuscript that identifies ribosome biogenesis and inhibitors of RNA polymerase as a novel therapeutic targets for addressing resistance to Ras targeted therapies. Using patient derived organoids the authors convincingly demonstrate that MEK inhibition results in upregulation of ribosome components yet reduced protein synthesis analogous to the phenotype of embryonic stem cells. Various experiments in cells demonstrate synergistic effects of Ras pathway + RNA polymerase inhibition that importantly are confirmed in a mouse model of colorectal cancer in Figure 3k. They also provide some interesting data correlating AMG510 sensitivity to ribosome biogenesis that may provide the basis for future triaging of patients for AMG510 treatment. The interpretation of the results and the discussion were fair and highlighted weaknesses and areas of future work. The entire manuscript was written and presented. I have a couple of minor comments:

1. Drug dosing - the authors need to provide context about whether the doses used are equivalent to those being used in clinical trials.
2. Provide more detail in Figure 3 legend about the genetics of the Ras driven colorectal cancer model.
3. Different Ras mutations are not all equally capable of driving oncogenesis. The authors have used KG12A and KG12C patient derived organoids and KG12D mice. Did they observe any differences in the strength of the phenotype or responsiveness with these different mutations? A comment in the discussion would be good.

Reviewer #2

(Remarks to the Author)

In the present manuscript Tanaka and coworkers investigate the effect of selected compounds inhibiting MAP-kinase and K-Ras on the behavior and gene expression profile of colon cancer patient derived organoids. They found that these inhibitors strongly decrease cellular protein synthetic activity and upregulate ribosome biogenesis. The use of the selective Pol I inhibitor CX-5461 showed a synergistic effect on patient derived organoids survival.

Overall this is a comprehensive manuscript using powerful experimental approaches, however, some additional analysis can strongly improve Authors' conclusions.

Major Points:

- In Fig. 1, 2 and 3 the ascribed affects of the treatments on ribosome biogenesis is only implied indirectly. Namely increased levels of a series of mRNA not necessarily implies that a cellular process is unregulated, particularly if global protein synthesis is reduced.

Experiments assessing the rate of ribosome production are missing. The evaluation of rRNA synthesis and/or of ribosomal protein expression (at the protein level) would strongly support Authors' conclusions.

- Transcriptomic and phosphoproteomic data (Fig. 1, 2) are never validated, even few representative top deregulated hits or the results of the subsequent functional characterization analysis should be tested individually by independent technical approaches. As an example, in Fig. 2 Authors identify a signature of deregulated proteins at the phosphorylation levels including NOTCH, leptin etc. Confirming this result by Western blot would be relatively easy and would strongly strengthen the findings.

Minor points:

Pag 2 line 82 POO should be PDO

Page 4 line 99, a mention to Extended Figure 1A should be added here

Figure 1 legend: panel G and H appear to be mislabeled. Grand state should spell ground state (check this throughout the manuscript).

Line 199-200. post-transcriptional should sound post-translational

line 230 refers to figure 3 b, which drug was used: trametinib or dinaciclib?

Reviewer #3

(Remarks to the Author)

Reviewer #4

(Remarks to the Author)

KRAS mutant CRC are a real challenge as these tumoral cells do not respond to KRAS-targeting therapeutic. The manuscript by Tanaka et al. describe the pathways that are relevant when KRAS mut CRC tumours are treated by the MEK inhibitor, trametinib and allele-specific KRAS inhibitor, AMG510. By using PDOs the authors suggest that ribosome biogenesis and Myc expression induced by these treatments are resembling the pluripotent naive ground state of ESCs. The authors also show data of how these inhibitors can act when co-treated with a RNA pol I inhibitor, CX-5461.

Major points:

Figure 1. It shows that a signature of myc-related ribosome biogenesis is induced when PDOs KRAS mut are being treated with trametinib. It is not convincing with these few experiments that is really resembling the pluripotent ground state.

Figure 2. Phospho-proteomics was done but results are broadly mentioned and further experiments done in organoids is a must.

Figure 3. Mouse experiments are not convincing and images quality is low and should be better described in the panel. The main question is if all the cells (healthy and cancer) are dying because of toxicity? the histology of both treatments in the intestine looks as fatal loss of intestinal homeostasis. Cancer cells are poorly detected even in all the images because of low quality.

Figure 4. Data shown is poor considering what was collected using this methodology. This also need to be improved to be convincing and also to include biological data with PDOs showing increased ribosome biogenesis.

Figure 5. Only one PDOs mutant was responding (1 out of 5?), what is happening to the non-responders? Synergistic effect data is only shown in 5e of this figure and the heatmaps do not show one treatment only, just both with different concentrations. This is not convincing and again points out that healthy cells are affected when both reagents are co-treated.

Abstract needs to be re-written to describe what the paper data is showing.

Title is also not representing the data shown.

Minor points:

1. lane 33 and 34, the sentence "KRAS targeting therapeutic agents are not very effective against CRC", this seems very general and more rigorous statement needs to be written.

2. lane 82, POOs instead of PDOs

3. lane 95, "previously reported CRC markers", it is not clear how was it done.

4. lane 100, what about the other clusters?

5. lane 141, very broad writing "CRC cells adopt an ...", as only some type of CRC cells.

6. lane 146, why first writing K-RAS mut and wt and after the HCT24-14 and HCT24-8? should always be called the same name.

7. lane 164, what type of cell lines?

8. lane 168, more information should be added about the hours that cells were treated.

9. lane 192, complete name of NPM1 and NCL.

10. lane 212 and 213, FDR and NES information to be added so it can be comparable to the ones mentioned before.

11. lane 241, "GOCC terms" quite general, can be written some details?

12. lane 256, PDOs were done here, so it can be written in the text.

13. lane 295, the clone HCT248P needs to be introduced if was not done before.

14. lane 330, only the inhibition was in Ras mutants.

15. More details about how RNAseq was done in methods.

16. Specify what is included in ENR medium.

17. lane 571, the "generated compound", looks like is a typo.

Version 1:

Reviewer comments:

Reviewer #1

(Remarks to the Author)

I have no further comments. The authors responded to all of my comments (Reviewer 1) with suitable edits made to the manuscript.

Reviewer #2

(Remarks to the Author)

The Authors responded to all my previous concerns, I don't have further requests.

Reviewer #3

(Remarks to the Author)

Reviewer #4

(Remarks to the Author)

I would like to thank the authors for their responses and the additional experiments conducted to address the queries. While the manuscript has been improved, there are still some concerns that have not been fully addressed.

Fig. 1

The statement that CRC cells resemble an embryo-ground state is not convincing. Are the analyses in Figure 1 conducted on HCT24-8 and HCT24-14, or on other CRC PDOs? How many were analyzed? This crucial information is missing. A significant conclusion is drawn from this analysis, but if it was performed on only one control and one RASmut PDO, the data would be insufficient to support such a claim. Additionally, while HCT191 cells were treated, only the Fib marker and ribosome biogenesis were assessed.

The gene analysis in Figure 1h is not convincing, are these differences statistically significant? The protein blots in Figure 1e also lack clarity. How many replicates were performed? Was there any quantification? Visually, the upregulation of PRL23 and RPL13A needs stronger validation.

Lastly, Figure 1i lacks any quantification visible or clear enough, making it an inadequate basis for drawing conclusions about gene regulation.

Similarly, the immunostainings in Figures 1j and 1l are not convincing. Can the authors provide representative images that correspond to the quantification shown below?

Fig. 2

The phospho-proteomics data has been improved in terms of signalling pathways; however, further refinements are needed to make the findings more convincing. Specifically, the data should more clearly demonstrate how trametinib (timing) affects ribosome biogenesis in both control and PDO mutant models. Experiments of treated cells (timing) are only shown in the PDO mutant.

Fig.3 and 4

The authors claim a synergistic effect of trametinib with dinaciclib and trametinib with CX-5461. However, they only test the PDO mutant and not the healthy control. Only Figures 3a and 3b address this statement for dinaciclib, and Figures 3e and 3f for CX-5461. The rest of the data focuses on single-compound treatments rather than their synergy, making the findings less novel. It remains possible that both compounds are generally toxic to all PDO types rather than specifically synergistic.

The synergy is more convincingly demonstrated in Figure 4, which presents in vivo data, but only for CX-5461—there is no in vivo data for dinaciclib. Were all wild-type and Apc-Ras-Lgr5 mice analyzed at day 5? Can the authors provide additional images of the combination-treated WT mice? Furthermore, could they examine embryonic-ground state markers in these in vivo experiments? It is unclear why the study focuses solely on villi size without assessing any markers related to ribosome biogenesis or the embryonic ground state, which would strengthen the conclusions.

Immunostainings have been improved.

Fig. 5 and 6

The experiments presented focus on another mutant, G12C, but only include ribosome biogenesis data and Fib stainings, which are insufficient to support a definitive conclusion. The data for the combination of CX-5461 and AMG510 is limited to Figures 6g and 6h

The abstract and title are still not clear.

Title: CRC cells adopt the embryo-ground state in response to KRAS inhibition

The embryo-ground state was observed on few PDOs RAS mut

The embryo-ground state data can be improved.

Abstract

"synergistic anti-tumor effect with trametinib and AMG510 in .. mouse model and human PDOs.." only was done in vivo with trametinib and CX.

"CRC cells adopt a cellular state.."

a few PDOs RAS mut were done not CRC cells in general. It is confusing including all CRC cells embryo-ground state was observed in few experiments.

"these observations demonstrated that CRC cells become tolerant to K-RAS inhibition by exploiting the cellular program of early embryogenesis.."

again too general writing about CRC cells, as only was observed in PDOs RAS mut.

Version 2:

Reviewer comments:

Reviewer #4

(Remarks to the Author)

I would like to thank the authors for their efforts in providing more convincing data. The manuscript has been improved.

Major Points:

1. The authors have confirmed that the scRNA-seq data presented is derived from a single PDO. This should be clearly stated in both the main text, methodology and the figure legend. In this revised version, the abstract and title have been updated. However, it is clear that additional data will be necessary in the future to robustly support the claim of embryo ground state phenotype acquisition.
2. The western blot analyses are now shown to be statistically significant, which I consider sufficient. It is ultimately up to the editor to decide whether this level of evidence is acceptable.
3. The in vivo data is more convincing; however, the previously raised concerns have been only partially addressed. Additional experiments would have helped strengthen the conclusions. For example, it is unclear why only a single marker, such as fibrillarin, was used in some of the experiments. Relying on a single marker can raise questions about the robustness of the findings.
4. The connection between ribosome biogenesis and colorectal cancer/PDOs is not novel in the field. While this study includes PDO experiments and in vivo data, the main message is now different to before in terms of novelty and significance.

Overall, I am happy to leave it to the editor to judge the originality of the work and whether the data presented is sufficient to support the findings.

We thank the reviewers for their comments and insights, which have helped improve the manuscript. Point-by-point rebuttal is presented below. Reviewers' comments are in BLACK, and authors' response are in BLUE. The revisions in the manuscript are indicated in RED.

Reviewers' comments:

Reviewer #1 (Remarks to the Author):

This is an excellent manuscript that identifies ribosome biogenesis and inhibitors of RNA polymerase as a novel therapeutic targets for addressing resistance to Ras targeted therapies. Using patient derived organoids the authors convincingly demonstrate that MEK inhibition results in upregulation of ribosome components yet reduced protein synthesis analogous to the phenotype of embryonic stem cells. Various experiments in cells demonstrate synergistic effects of Ras pathway + RNA polymerase inhibition that importantly are confirmed in a mouse model of colorectal cancer in Figure 3k. They also provide some interesting data correlating AMG510 sensitivity to ribosome biogenesis that may provide the basis for future triaging of patients for AMG510 treatment. The interpretation of the results and the discussion were fair and highlighted weaknesses and areas of future work. The entire manuscript was written and presented. I have a couple of minor comments:

We thank the reviewer for recognizing the significance of our work.

1. Drug dosing - the authors need to provide context about whether the doses used are equivalent to those being used in clinical trials.

We thank the reviewer for this insightful comment. We used a dose of 0.1mg/kg of CX-5461 to evaluate the combination effects with trametinib, which is a significantly lower concentration compared to the previously reported 35-50mg/kg used to assess the efficacy of CX-5461 alone^{1,2}. According to the reviewer's suggestion, we calculated human equivalent dose (HED) using the interspecies allometric scaling for dose conversion³, and found it to be equivalent to 0.28 mg/m². In the first-in-human dose-escalation study of CX-5461, patients were treated at doses ranging from 25 to 450 mg/m²⁴. In the multicenter phase I trial involving patients with solid tumors, doses ranged from 50 to 650 mg/m²⁵. Thus, it appears that the estimated HED are lower than those being used in clinical trials, suggesting that a low dose of CX-5461 is sufficient for the combination therapy with trametinib. Although allometric scaling is an empirical approach that assumes unique characteristics in anatomical and physiological processes among species⁶, we believe that these results strengthen our claim that CX-5461 is a potential candidate for combination therapy with KRAS inhibition.

In the revised manuscript, we added these points in the discussion (page 11 line 338-345)

2. Provide more detail in Figure 3 legend about the genetics of the Ras driven colorectal cancer model.

We thank the reviewer for pointing out the insufficient description. In response to Reviewer 4, figures related to the mouse model of colorectal cancer was moved to Fig. 4. Mouse genotypes were explicitly shown in the figures, and the genetics was described in the figure legends. They are conditional knockout mice, and the detailed experimental procedure for introducing mutations is described in materials and methods (page 22 line 676-678).

3. Different Ras mutations are not all equally capable of driving oncogenesis. The authors have used KG12A and KG12C patient derived organoids and KG12D mice. Did they observe any differences in the strength of the phenotype or responsiveness with these different mutations? A comment in the discussion would be good.

We agree with the reviewer that different RAS mutations exhibit varying strength of the phenotype and the responsiveness against chemotherapeutic agents. This is exemplified by KRAS G13D mutation, which has been clinically proven to be sensitive to cetuximab⁷. Two proteomic studies reported mutation specific phosphorylation signature between KRAS G12D and G13D^{8,9}. However, in this study, we did not directly compare the differences among the distinct mutations. In the context of the recent development of mutation-specific and pan-mutant KRAS inhibitors, it is of particular interests to clarify the different response among the mutations. We believe that establishing organoids

with different KRAS mutations and conducting comparative analyses will be highly important experiment in the future.

As the reviewer suggested, we included these points in the discussion (page16 line483-491).

Reviewer #2 (Remarks to the Author):

In the present manuscript Tanaka and coworkers investigate the effect of selected compounds inhibiting MAP-kinase and K-Ras on the behavior and gene expression profile of colon cancer patient derived organoids. They found that these inhibitors strongly decrease cellular protein synthetic activity and upregulate ribosome biogenesis. The use of the selective Pol I inhibitor CX-5461 showed a synergistic effect on patient derived organoids survival.

Overall this is a comprehensive manuscript using powerful experimental approaches, however, some additional analysis can strongly improve Authors' conclusions.

We thank the reviewer for the overall positive feedback.

Major Points:

- In Fig. 1, 2 and 3 the ascribed affects of the treatments on ribosome biogenesis is only implied indirectly. Namely increased levels of a series of mRNA not necessarily implies that a cellular process is unregulated, particularly if global protein synthesis is reduced.

Experiments assessing the rate of ribosome production are missing. The evaluation of rRNA synthesis and/or of ribosomal protein expression (at the protein level) would strongly support Authors' conclusions.

Thank you for pointing out such highly relevant and important points, which we have followed to strengthen our conclusion of ribosome biogenesis. In Fig. 1, we evaluated the protein levels of ribosomal genes by immunoblot analysis, and demonstrated increased protein levels of genes identified through GOCC enrichment analysis following trametinib treatment (new Fig. 1e, page4 line118-121). According to the reviewer's suggestion, we have performed 5-ethylnyluridine (EU) labeling to detect newly transcribed RNA, and this new analysis clearly show that CX-5461 reduces rRNA synthesis in CRC organoids, as shown by substantial decrease of EU signal in the nucleoli (new Fig. 3i and j, page10 line300-303). Overall, we believe these additional data validated the RNA-seq data.

- Transcriptomic and phosphoproteomic data (Fig. 1, 2) are never validated, even few representative top deregulated hits or the results of the subsequent functional characterization analysis should be tested individually by independent technical approaches. As an example, in Fig. 2 Authors identify a signature of deregulated proteins at the phosphorylation levels including NOTCH, leptin etc. Confirming this result by Western blot would be relatively easy and would strongly strengthen the findings.

We validated the transcriptomic data of three representative up-regulated ribosome protein via immunoblot as mentioned above (new Fig. 1e). Among 8354 phosphorylation sites identified in proteome analysis, only 489 sites have shown functional evidences. We assessed the phosphorylation state of ribosomal protein S6, a commonly used read out of mTORC1 activity, because it is a major regulator of protein synthesis. Consistent with the observation indicating the reduced protein synthesis following trametinib treatment (Fig. 1k), immunoblot analysis showed significant reduction of Ser235/Ser236 and Ser240/244 confirming the phosphoproteomic data. Additionally, key regulator of cell proliferation, Ser807/Ser811 of RB1 and Ser727 of STAT1, as well as primary mediator of transcriptional response to proteotoxic response, Ser303/Ser307 of HSF1, were decreased, further confirming the phosphoproteomic data. Unfortunately, no antibodies against phosphorylation sites related to NOTCH or leptin were available. However, we believe that the consistency between the phosphorylation analysis by immunoblot and the phosphoproteome analysis validate our findings. The results of immunoblot and the heatmap showing phosphoproteomic data of corresponding phosphorylation sites were shown in Extended Data Fig. 2e and f, respectively, and text was amended accordingly (page8 line 245-page9 line256).

Minor points:

Pag 2 line 82 POO should be PDO

Thank you for pointing out this typo. We amended the spelling.

Page 4 line 99, a mention to Extended Figure 1A should be added here

Text was amended.

Figure 1 legend: panel G and H appear to be mislabeled. Grand state should spell ground state (check this throughout the manuscript).

Thank you for pointing out the mislabel. We fixed them in fig. 1j and k. We have confirmed that the spelling of “ground state” is consistent throughout the manuscript.

Line 199-200. post-transcriptional should sound post-translational

Thank you for pointing out the important mistake. We have corrected it.

line 230 refers to figure 3 b, which drug was used: trametinib or dinaciclib?

Thank you again for pointing out the important mistake. We have collected ‘trametinib’ to ‘dinaciclib’.

Reviewer #3 (Remarks to the Author):

Reviewer #4 (Remarks to the Author):

KRAS mutant CRC are a real challenge as these tumoral cells do not respond to KRAS-targeting therapeutic. The manuscript by Tanaka et al. describe the pathways that are relevant when KRAS mut CRC tumours are treated by the MEK inhibitor, trametinib and allele-specific KRAS inhibitor, AMG510. By using PDOs the authors suggest that ribosome biogenesis and Myc expression induced by these treatments are resembling the pluripotent naive ground state of ESCs. The authors also show data of how these inhibitors can act when co-treated with a RNA pol I inhibitor, CX-5461.

We would like to thank the reviewer for critical reading of our manuscript.

Major points:

Figure 1. It shows that a signature of myc-related ribosome biogenesis is induced when PDOs KRAS mut are being treated with trametinib. It is not convincing with these few experiments that is really resembling the pluripotent ground state.

We thank the reviewer for raising this critical point. We did not make any claims as to whether CRC cells have acquired pluripotency after trametinib treatment. We argue that there is a similarity in the gene expression profile between trametinib-treated CRC organoids and ground-state ESCs and epiblasts. In addition, the accumulation of fibrillar in the nucleolus and the decrease in OPP incorporation reproduced two hallmark features of the ground state: enhanced ribosome biogenesis and decreased global protein synthesis. Based on these observations, we suggested the similarities in ribosome biogenesis and translation, but did not assert the pluripotency. We revised the text to avoid expressions that might give the misleading impression that CRC cells could acquire pluripotency (page4 line123, page4 line125 and page12 line371).

In response to the reviewer’s concern, we further evaluated the cell state using two approaches. First, we analyzed the expression of genes that were significantly upregulated in the mouse ground state epiblast. In line with our hypothesis, these genes were confirmed to be significantly highly expressed in C1 ($p_{val_adj} < 0.05$) (new Fig. 1h, page5 line 141-143). Interestingly, this analysis identified three genes, including SLC12A2, PDGFA and RPL34, were also found to be significantly highly expressed in C5 ($p_{val_adj} < 0.05$, page5 line143-147). Considering that C5 is absent in trametinib-free medium and is induced by trametinib treatment, these new gene-based analysis provided additional evidence indicating the similarity between trametinib-induced cell state of CRC and ground state epiblast. Based on these results, we calculated the gene score for ribosome biogenesis feature, and found C1 and C5 exhibited higher score compared to other clusters (new Extended Fig. 1e, page5 line 147-148).

Secondly, we focused on ribosome-related genes that are highly expressed in human ESC^{10,11}. The heatmap demonstrated that they were highly expressed in C1 (new Fig. 1i, page5 line149-157). We noticed that RPL13A was also upregulated in trametinib treated fraction of C3. We believe these comparative analyses of human ESCs further support the notion of the similarity in the cell state between trametinib treated CRC organoids and the ground state epiblast. Additionally, we demonstrated that trametinib reduced the global protein synthesis (Fig. 1k) and reduced cell proliferation rate (Fig. 1l), which are hallmark of pre-implanted epiblast. We believe these observations demonstrate the significant similarity between trametinib-induced cell state of CRC organoids and the ground state embryonic cells.

Figure 2. Phospho-proteomics was done but results are broadly mentioned and further experiments done in organoids is a must.

According to the reviewer's suggestion, we analyzed the phosphorylation state of key molecules involved in mTOR signaling, because it plays central roles in the protein translation. PTMsigDB analysis identified two KINASE-PSP pathways, p90RSK/RPS6KA3 and RSK2/RPS6KA3, as negatively enriched pathways in response to trametinib treatment (Fig. 2h). The phosphorylation of Ser235/Ser236 and Ser240/Ser244 of RPS6, a commonly used as a readout of the mTOR signaling, were negatively enriched in trametinib treated CRC organoids in phospho-proteomics analysis (new Extended Data Fig. 2e). Additionally, the phosphorylation of two mTOR-linked molecules including HSF1 Ser303/Ser307 and STAT1 S727 were also negatively enriched^{12,13}. The reduced phosphorylation of these molecules was validated with immunoblot analysis using phospho-specific antibodies (new Extended Data Fig. 2f). These results explain the reduced global protein synthesis after trametinib treatment. We described these results in the text (page8 line240-page9 line256)

Figure 3. Mouse experiments are not convincing and images quality is low and should be better described in the panel. The main question is if all the cells (healthy and cancer) are dying because of toxicity? the histology of both treatments in the intestine looks as fatal loss of intestinal homeostasis. Cancer cells are poorly detected even in all the images because of low quality.

Thank you again for the critical comments. We apologize the low quality of the provided images in the manuscript, which is due to the significant deterioration of the images when they were consolidated into a single file during the initial submission. In the revised manuscript, we reorganized mouse experiments data into a new standalone Figure 4. Each panel was enlarged to create a clear and more comprehensive illustration. In immunofluorescence images, tumor cells were identified by CD44v6 (Fig. 4a and e) or β -catenin staining (Fig. 4c and g). The nucleolar staining of fibrillarlin in tumor cells is shown in the insets (Fig. 4a).

In the initial submission, we only analyzed tumor tissues, because the primary aim of this work is to present a finding on the previously unrecognized cellular state of intestinal tumor cells after RAS inhibition. However, we do agree that the toxic effects to healthy cells is a critical issue to consider for clinical applications, so we have tried to address this issue as much as possible. We additionally

performed the immunofluorescence analysis of normal tissues, and found trametinib treatment does not significantly altered either fibrillar intensity (new Fig. 4e and f) or EdU incorporation (new Fig. 4g and h). These results may support the notion that normal tissues are less sensitive to trametinib treatment than tumor tissue. We evaluated the effects of trametinib and CX-5461 by measuring size of villi (new Fig. 4k). CX-5461 did not significantly affected, but trametinib and combination of trametinib and CX-5461 reduced the size by 80.8% and 79.7%, respectively (new Fig. 4l). Thus, trametinib monotherapy exhibit inhibitory effects on normal tissue, but no synergistic effect with CX-5461 was observed. Importantly, the combination of trametinib and CX-5461 demonstrates significantly higher inhibitory effects on tumor tissues compared to normal tissue. We believe these results support the effectiveness of the combination of trametinib an CX-5461 as a novel therapeutic strategy for CRC (page 11 line 332-335).

Figure 4. Data shown is poor considering what was collected using this methodology. This also need to be improved to be convincing and also to include biological data with PDOs showing increased ribosome biogenesis.

In response to the reviewer's criticism, we further analyzed scRNA-seq data. We included the proportion plot to highlight that the percentage of C2 cluster in HCT191-1T is higher than in other organoids (new Fig. 5c). The heatmap of genes identified in GOCC enrichment analysis clearly indicated the increased expression of ribosomal protein genes in HCT191-1T (new Fig. 5e). Furthermore, we calculated the gene score for ribosome biogenesis in each cell and, C2 showed a high value (new Fig. 5f). Interestingly, this analysis newly revealed that ribosome biogenesis is also enhanced in C5. Considering that the proportion of C5 is higher in HCT191-1T compared to other organoids (new Fig. 5c), these findings support the correlation between ribosome biogenesis and

response to AMG510 (page12 line362-368).

According to the reviewer's suggestion, we performed two biological experiments. 5-ethyluridine (EU) labeling detects increased transcription of ribosome DNA in HCT191-1T (new Fig. 5i and j). Fibrillar staining indicated its increased accumulation at nucleoli (Fig. 5k and l). Both experiments demonstrated that HCT191-1T not only exhibit an increase expression of genes encoding ribosomal protein, but also enhance ribosome RNA expression and its modification. We believe these additional data further support our notion that HCT191-1T increased ribosome biogenesis (page12 line377-page13 line383).

Figure 5. Only one PDOs mutant was responding (1 out of 5?), what is happening to the non-responders? Synergistic effect data is only shown in 5e of this figure and the heatmaps do not show one treatment only, just both with different concentrations. This is not convincing and again points out that healthy cells are affected when both reagents are co-treated.

As noted by the reviewer, only one PDO responded to AMG510. In the first clinical trial of sotorasib, a partial response was observed in three out of 42 CRC patients¹⁴. More recently, the objective response was observed in six out of 62 CRC patients¹⁵. Thus, the response rate of PDOs appears to reflect the efficacy of clinical trials.

Non-responders exhibited reduced EdU incorporation, but the reduction was less pronounced compared to responders (Fig. 6a). This difference is thought to reflect the distinct responses of CRC organoids to AMG510. Two hallmarks of ground state ESCs, including the increased transcription of ribosome genes (Fig. 6d) and the decreased global protein synthesis (Fig. 6b), were observed in the non-responders. To further evaluate the response of non-responders, we performed fibrillar staining

and found the elevated intensity (new Fig. 6c and new Extended data Fig. 6c). Based on these observations, we concluded that AMG510 induced KRAS G12C-mutant CRC organoids into a cellular state resembling ground-state ESCs, mirroring the effect observed in KRAS-mutant organoids treated with trametinib, as shown in Fig. 1 (page13 line398-401).

According to the reviewer's comment, we have added the dose-effect curve for CX-5461 alone (new Fig. 6f) and mentioned in the text (page 14 line 413-414). The dose-effect curves for AMG510 is shown in Figure 4a. Chou-Talalay methods is widely used in the evaluation of chemotherapeutic agents¹⁶. We presented the results by Fa-CI plot for the constant ratio evaluation and by heatmap for the non-constant ratio evaluation, because this method is based on the median-effect equation and provides common link between single entity and multiple entities, respectively.

To establish organoids, we adopted the protocol developed by Sato and Cleavers group, which have been widely used for establishing PDOs^{17,18}. This culture condition is designed to exclude the growth of normal colorectal cells, ensuring that the readout of this analysis specifically represented the response of tumor cells. Unfortunately, normal organoids have not been established from these patients.

Fig. 6. c. Increased fibrillar expression in AMG510-treated KRAS G12C CRC organoids. The intensity of fibrillar signals in each nuclei per organoids is shown in the boxplot. Data of 20 organoids obtained from four independent experiments are shown. **p < 0.01, *p < 0.05. **f.** Response of KRAS G12C organoids to CX-5461. Cell viabilities of organoids were evaluated at 72 hours after treatment with indicated concentrations of CX-5461. Mean ± standard deviation from four independent experiments is shown.

Extended Fig. 6c. Fibrillar expression was visualized by immune fluorescent staining (c). Bar = 50 μm.

Abstract needs to be re-written to describe what the paper data is showing.

Title is also not representing the data shown.

We appreciate the feedback.

We substantially edited the abstract to improve the clarity of our findings. Title was revised to more accurately reflect the content of the manuscript.

Minor points:

1. line 33 and 34, the sentence "KRAS targeting therapeutic agents are not very effective against CRC", this seems very general and more rigorous statement needs to be written.

To provide a more specific description of the response rate for CRC, we cited the report of phase II

clinical trial and included information on ORR and PFS as follows:

“A phase 2 clinical trial of KRAS G12C allele specific inhibitor, AMG510, in colorectal cancer (CRC) reported an objective response rate of 9.7% and a median progression-free survival of 4.0 months¹⁵”

2. lane 82, POOs instead of PDOs

Amended.

3. lane 95, "previously reported CRC markers", it is not clear how was it done.

We amended the sentence as below.

“Each cluster was initially annotated by calculating the p-value of overlaps using Fisher’s exact test between the differentially expressed genes (DEGs) of each cluster and CRC markers reported previously”.

4. lane 100, what about the other clusters?

We added following sentences.

“C3 and C4 were annotated as absorptive cells and transit-amplifying cells, respectively, with their proportions remaining relatively unchanged. C5 was annotated as goblet cells, which were absent under untreated conditions but was induced following trametinib treatment.”

5. lane 141, very broad writing "CRC cells adopt an ...", as only some type of CRC cells.

We amended the sentence as follows.

“Collectively, these observations suggest that some trametinib-tolerant KRAS mutant CRC cells are induced into an embryonic ground-state-like cellular state by trametinib treatment.”

6. lane 146, why first writing K-RAS mut and wt and after the HCT24-14 and HCT24-8? should always be called the same name.

We have corrected the points and standardized the terminology.

7. lane 164, what type of cell lines?

They are isogenic established cell lines with various KRAS mutations. Specific cell names and genetic mutations have been included.

8. lane 168, more information should be added about the hours that cells were treated.

The treatment time points have been included.

9. lane 192, complete name of NPM1 and NCL.

Complete names have been included.

10. lane 212 and 213, FDR and NES information to be added so it can be comparable to the ones mentioned before.

FDR and NES have been included.

11.lane 241, "GOCC terms" quite general, can be written some details?

GOCC (Gene ontology cellular components) is explained at its first mention on page 4 line110.

12. lane 256, PDOs were done here, so it can be written in the text.

We added PDOs into the relevant section.

13. lane 295, the clone HCT248P needs to be introduced if was not done before.

This is a typo for HCT24-8. We have corrected it.

14. lane 330, only the inhibition was in Ras mutants.

We have added “in KRAS-mutant CRC organoids” to the relevant sentence.

15. More details about how RNAseq was done in methods.
The details of RNA-seq analysis have been described (page19 line576-585)

16. Specify what is included in ENR medium.
ENR has been explained in the first instance on page17 line528.

17. lane 571, the "generated compound", looks like is a typo.
This is not typo, but it has been corrected as follows to more accurately represent the genotype of the mutant mouse.

“The generated *Apc580^{SS}*, *Kras^{G12D/+}* double conditional knockouts bearing *Lgr5-CreERT2*, *Apc580^{SS}*, *Kras^{G12D/+}* were crossed with a homozygous background of *Apc580^{SS}* bearing *Lgr5-CreERT2*.”

References for point-by-point response

- 1 Kusnadi, E. P. *et al.* Reprogrammed mRNA translation drives resistance to therapeutic targeting of ribosome biogenesis. *EMBO J* **39**, e105111 (2020). <https://doi.org/10.15252/emj.2020105111>
- 2 Otto, C. *et al.* RNA polymerase I inhibition induces terminal differentiation, growth arrest, and vulnerability to senolytics in colorectal cancer cells. *Mol Oncol* **16**, 2788-2809 (2022). <https://doi.org/10.1002/1878-0261.13265>
- 3 Nair, A. B. & Jacob, S. A simple practice guide for dose conversion between animals and human. *J Basic Clin Pharm* **7**, 27-31 (2016). <https://doi.org/10.4103/0976-0105.177703>
- 4 Khot, A. *et al.* First-in-Human RNA Polymerase I Transcription Inhibitor CX-5461 in Patients with Advanced Hematologic Cancers: Results of a Phase I Dose-Escalation Study. *Cancer Discov* **9**, 1036-1049 (2019). <https://doi.org/10.1158/2159-8290.CD-18-1455>
- 5 Hilton, J. *et al.* Results of the phase I CCTG IND.231 trial of CX-5461 in patients with advanced solid tumors enriched for DNA-repair deficiencies. *Nature communications* **13**, 3607 (2022). <https://doi.org/10.1038/s41467-022-31199-2>
- 6 Chaturvedi, P. R., Decker, C. J. & Odinecs, A. Prediction of pharmacokinetic properties using experimental approaches during early drug discovery. *Curr Opin Chem Biol* **5**, 452-463 (2001). [https://doi.org/10.1016/s1367-5931\(00\)00228-3](https://doi.org/10.1016/s1367-5931(00)00228-3)
- 7 De Roock, W. *et al.* Association of KRAS p.G13D mutation with outcome in patients with chemotherapy-refractory metastatic colorectal cancer treated with cetuximab. *JAMA* **304**, 1812-1820 (2010). <https://doi.org/10.1001/jama.2010.1535>
- 8 Hammond, D. E. *et al.* Differential reprogramming of isogenic colorectal cancer cells by distinct activating KRAS mutations. *J Proteome Res* **14**, 1535-1546 (2015). <https://doi.org/10.1021/pr501191a>
- 9 Tahir, R. *et al.* Mutation-Specific and Common Phosphotyrosine Signatures of KRAS G12D and G13D Alleles. *J Proteome Res* **20**, 670-683 (2021). <https://doi.org/10.1021/acs.jproteome.0c00587>
- 10 Corsini, N. S. *et al.* Coordinated Control of mRNA and rRNA Processing Controls

- Embryonic Stem Cell Pluripotency and Differentiation. *Cell stem cell* **22**, 543-558 e512 (2018). <https://doi.org/10.1016/j.stem.2018.03.002>
- 11 Richards, M., Tan, S. P., Tan, J. H., Chan, W. K. & Bongso, A. The transcriptome profile of human embryonic stem cells as defined by SAGE. *Stem cells* **22**, 51-64 (2004). <https://doi.org/10.1634/stemcells.22-1-51>
- 12 Santagata, S. *et al.* Tight coordination of protein translation and HSF1 activation supports the anabolic malignant state. *Science* **341**, 1238303 (2013). <https://doi.org/10.1126/science.1238303>
- 13 Shamovsky, I., Ivannikov, M., Kandel, E. S., Gershon, D. & Nudler, E. RNA-mediated response to heat shock in mammalian cells. *Nature* **440**, 556-560 (2006). <https://doi.org/10.1038/nature04518>
- 14 Hong, D. S. *et al.* KRAS(G12C) Inhibition with Sotorasib in Advanced Solid Tumors. *N Engl J Med* **383**, 1207-1217 (2020). <https://doi.org/10.1056/NEJMoa1917239>
- 15 Fakhri, M. G. *et al.* Sotorasib for previously treated colorectal cancers with KRAS(G12C) mutation (CodeBreak100): a prespecified analysis of a single-arm, phase 2 trial. *Lancet Oncol* **23**, 115-124 (2022). [https://doi.org/10.1016/S1470-2045\(21\)00605-7](https://doi.org/10.1016/S1470-2045(21)00605-7)
- 16 Chou, T. C. Drug combination studies and their synergy quantification using the Chou-Talalay method. *Cancer Res* **70**, 440-446 (2010). <https://doi.org/10.1158/0008-5472.CAN-09-1947>
- 17 Sato, T. *et al.* Single Lgr5 stem cells build crypt-villus structures in vitro without a mesenchymal niche. *Nature* **459**, 262-265 (2009). <https://doi.org/10.1038/nature07935>
- 18 Sato, T. *et al.* Long-term expansion of epithelial organoids from human colon, adenoma, adenocarcinoma, and Barrett's epithelium. *Gastroenterology* **141**, 1762-1772 (2011). <https://doi.org/10.1053/j.gastro.2011.07.050>

We thank the reviewers for their comments and insights, which have helped improve the manuscript. Point-by-point rebuttal is presented below. Reviewers' comments are in BLACK, and authors' response are in BLUE. The revisions in the manuscript are indicated in RED.

REVIEWER COMMENTS

Reviewer #1 (Remarks to the Author):

I have no further comments. The authors responded to all of my comments (Reviewer 1) with suitable edits made to the manuscript.

Reviewer #2 (Remarks to the Author):

The Authors responded to all my previous concerns, I don't have further requests.

Reviewer #3 (Remarks to the Author):

Reviewer #4 (Remarks to the Author):

I would like to thank the authors for their responses and the additional experiments conducted to address the queries. While the manuscript has been improved, there are still some concerns that have not been fully addressed.

We would like to sincerely thank reviewer #4 for the thorough and thoughtful review.

Fig. 1

The statement that CRC cells resemble an embryo-ground state is not convincing. Are

the analyses in Figure 1 conducted on HCT24-8 and HCT24-14, or on other CRC PDOs? How many were analyzed? This crucial information is missing. A significant conclusion is drawn from this analysis, but if it was performed on only one control and one RASmut PDO, the data would be insufficient to support such a claim. Additionally, while HCT191 cells were treated, only the Fib marker and ribosome biogenesis were assessed.

Given the substantial expense of scRNA-seq, we limited the analysis to HCT24-8. In Figure 6, we performed bulk RNA-seq on four PDOs and confirmed that KRAS inhibition led to increased Myc-dependent transcriptional activity. Furthermore, enrichment of the ribosome signature was observed in KRAS inhibitor-tolerant clones after drug treatment, supporting our conclusions.

Nevertheless, we understand the reviewer's concern regarding the limited number of samples in this figure. Therefore, we have avoided to generalize the findings from this analysis. We also revised the conclusions regarding the embryonic ground state and amended the text as follows (page6, line 171-173) :

Collectively, these observations suggest that trametinib induces a cellular state in KRAS-mutant colorectal cancer PDOs in which Myc-dependent ribosome biogenesis is upregulated.

The gene analysis in Figure 1h is not convincing, are these differences statistically significant? The protein blots in Figure 1e also lack clarity. How many replicates were performed? Was there any quantification? Visually, the upregulation of PRL23 and RPL13A needs stronger validation.

In response to the reviewer's comments, we have included the statistical analysis results for Figure 1h in supplementary Table1, which indicate a significantly different expression of all genes in cluster C1 (p_val_adj ranging from 2.91E-198 to 0.00031). The immunoblot presented in Figure 1e was conducted in triplicate, and the corresponding quantification results are described in the text (page4, line118-121).

Lastly, Figure 1i lacks any quantification visible or clear enough, making it an inadequate basis for drawing conclusions about gene regulation.

In Figure 1i, we showed the result as a heatmap because it is commonly used to present the quantification data of gene expression in single-cell analysis. In revision, we have added the statistical data for this figure as a supplementary table 2 to demonstrate the validity of our conclusion from this analysis (adj_p_val < 1e10⁻⁶²).

Similarly, the immunostainings in Figures 1j and 1l are not convincing. Can the authors provide representative images that correspond to the quantification shown below?

In accordance with the reviewer's suggestion, we have provided clear representative images by excluding the bright-field images.

Fig. 2

The phospho-proteomics data has been improved in terms of signalling pathways; however, further refinements are needed to make the findings more convincing. Specifically, the data should more clearly demonstrate how trametinib (timing) affects ribosome biogenesis in both control and PDO mutant models. Experiments of treated cells (timing) are only shown in the PDO mutant.

The response of wild-type PDOs to trametinib is certainly intriguing. However, the phospho-proteomics analysis in the KRAS wild-type PDOs are technically difficult, as they are eradicated by trametinib treatment, making it extremely difficult to distinguish between the response to KRAS inhibition and the secondary response associated with cell death. Accordingly, trametinib-induced changes were analyzed exclusively in KRAS-mutant PDOs, and no in wild-type ones.

Fig.3 and 4

The authors claim a synergistic effect of trametinib with dinaciclib and trametinib with CX-5461. However, they only test the PDO mutant and not the healthy control. Only Figures 3a and 3b address this statement for dinaciclib, and Figures 3e and 3f for CX-5461. The rest of the data focuses on single-compound treatments rather than their synergy, making the findings less novel. It remains possible that both compounds are generally toxic to all PDO types rather than specifically synergistic.

We appreciate the reviewer for highlighting this important point. While the response of normal cells is indeed an important issue, we have not established PDOs from normal tissue. To address this point, we have included a new analysis of wild-type mice in Figure 4.

To the best of our knowledge, this is the first report of a synergistic effect of dinaciclib or CX-5461 with trametinib. Therefore, we believe that elucidating the pharmacological action of each drug as single agent against CRC is of great significance.

We acknowledge the possibility that the observed effects may be due to general toxicity. However, our combination index (CI) analysis demonstrated CI values consistently <1 ,

suggesting a synergistic interaction rather than non-specific toxicity. To strengthen our understanding of the synergistic interaction, we analyzed the median-effect principle. The resulting plot clearly demonstrated that the effect of each drug- both as single agent and in combination of trametinib- are consistent with the mass-action law. These new results were shown as Supplemental Data Figure 3a and b, and described in page 9, line272-279 and page 10, line 317-319. We believe these results, together with the previously presented combination index data, support that dinaciclib and CX-5461 exhibit synergistic effect when combined with trametinib.

The synergy is more convincingly demonstrated in Figure 4, which presents *in vivo* data, but only for CX-5461—there is no *in vivo* data for dinaciclib. Were all wild-type and Apc-Ras-Lgr5 mice analyzed at day 5? Can the authors provide additional images of the combination-treated WT mice? Furthermore, could they examine embryonic-ground state markers in these *in vivo* experiments? It is unclear why the study focuses solely on villi size without assessing any markers related to ribosome biogenesis or the embryonic ground state, which would strengthen the conclusions.

We did not perform *in vivo* analysis of dinaciclib, as it shows limited synergy with the KRAS inhibitor and affects pathways beyond ribosome biogenesis, rendering it unsuitable for further evaluation of synergistic effects with KRAS inhibition.

In response to the reviewer's comments regarding the experimental protocol, we have provided the illustration of the experimental timeline in Figure 4a and f. To analyze

ribosome biogenesis, we examined fibrillarin expression six hours after a single administration of trametinib. To evaluate the antitumor effects of the drug treatment, we performed the analysis 24 hours after the fifth administration. Additional images of the combination-treated WT mice were included in Figure 4k and m. The text has been amended accordingly (page 11, line 330-351).

We have shown expression of fibrillarin to demonstrate the elevated ribosome biogenesis after trametinib treatment. As the reviewer pointed out, evaluation of ribosome biogenesis does refine our conclusions; however, establishing pathological markers requires considerable time and efforts. Therefore, we would like to leave this as a task for future studies.

Immunostainings have been improved.

We appreciate the favorable evaluation of the immunostainings.

Fig. 5 and 6

The experiments presented focus on another mutant, G12C, but only include ribosome biogenesis data and Fib stainings, which are insufficient to support a definitive conclusion. The data for the combination of CX-5461 and AMG510 is limited to Figures 6g and 6h

We respectfully disagree with the reviewer's comment that the data presented in Figure 5 and 6 are insufficient. The GOCC and MSigDB gene sets that identified through comprehensive analyses of bulk and scRNA-seq data consistently exhibited extremely high NES scores. The ground state gene signature of ESCs and epiblast are also significantly enriched. We believe these results strongly support our conclusions.

We consider Figures 6g and 6h to provide sufficient evidence for the synergistic effect between CX-5461 and AMG510. Additionally, the enrichment of ribosome biogenesis and the suppression of protein synthesis are demonstrated in Figure 6a-e, respectively. In response to the reviewer's concern, we performed an additional EU incorporation assay, which demonstrated that CX-5461 reduces nascent RNA synthesis. The results have been shown in Extended Data Fig. 6d and the text has been amended accordingly (Page 14, line 430-431). We believe these results provide sufficient evidence to support our conclusions.

The abstract and title are still not clear.

Title: CRC cells adopt the embryo-ground state in response to KRAS inhibition

The embryo-ground state was observed on few PDOs RAS mut

The embryo-ground state data can be improved.

In accordance with the reviewer's comments, we tone down conclusions on the embryo-ground state and focus on the potential of ribosome biogenesis as a therapeutic target in KRAS-mutant colorectal cancer. The revised title is as follows:

'Ribosome biogenesis as a potential therapeutic target in KRAS mutant colorectal cancer'.

Abstract

We have made substantial revision to the abstract.

"synergistic anti-tumor effect with trametinib and AMG510 in .. mouse model and human PDOs.."

only was done in vivo with trametinib and CX.

According to the reviewer's suggestion, we have amended the text.

We found that they were vulnerable to the inhibition of RNA polymerase I, and it exhibited synergistic anti-tumor effects with trametinib in an autochthonous mouse model of intestinal tumors and human patient-derived organoids (PDOs). (Page2, line45-48)

"CRC cells adopt a cellular state.."

a few PDOs RAS mut were done not CRC cells in general. It is confusing including all CRC cells

embryo-ground state was observed in few experiments.

According to the reviewer's suggestion, we have removed the term 'embryonic ground state', and revised this paragraph as follows:

Here, we demonstrated that a subset of KRAS-mutant CRC cells transitions to a cellular state characterized by enhanced ribosome biogenesis upon KRAS signaling inhibition. (Page2, line40-42)

"these observations demonstrated that CRC cells become tolerant to K-RAS inhibition by exploiting the cellular program of early embryogenesis.."

again too general writing about CRC cells, as only was observed in PDOS RAS mut.

We have modified the text according to the reviewer's suggestion.

These observations demonstrated that high ribosome biogenesis induced by KRAS inhibition is indispensable to maintain this cellular state and is a potential therapeutic target. (Page2, line48-52)

Reviewer #4 (Remarks to the Author):

I would like to thank the authors for their efforts in providing more convincing data. The manuscript has been improved.

We sincerely appreciate your positive feedback and are glad to hear that the revisions have improved the manuscript.

Major Points:

1. The authors have confirmed that the scRNA-seq data presented is derived from a single PDO. This should be clearly stated in both the main text, methodology and the figure legend. In this revised version, the abstract and title have been updated. However, It is clear that additional data will be necessary in the future to robustly support the claim of embryo ground state phenotype acquisition.

In accordance with the reviewer's suggestions, we have revised the text and figure legends to clearly state that the analysis was conducted using a single PDO (page 3, line88 and figure 1a legend). In this study, we performed scRNA-seq analysis on a total of five PDOs, with the detailed methodology provided in the "Single-cell RNA sequencing" section.

2. The western blot analyses are now shown to be statistically significant, which I consider sufficient. It is ultimately up to the editor to decide whether this level of evidence is acceptable.

We appreciate the favorable evaluation of the western blot. We have prepared the manuscript in accordance with the submission guidelines of *Nature Communications*.

3. The *in vivo* data is more convincing; however, the previously raised concerns have been only partially addressed. Additional experiments would have helped strengthen the conclusions. For example, it is unclear why only a single marker, such as fibrillarin, was used in some of the experiments. Relying on a single marker can raise questions about the robustness of the findings.

We appreciate your further comments. We consider that this *in vivo* analysis supports the conclusions drawn from the scRNA-seq results. Nevertheless, as you have rightly noted, the evidence relies on fibrillarin staining, and we will address this point in future studies by elucidating the molecular mechanisms underlying ribosome biogenesis.

4. The connection between ribosome biogenesis and colorectal cancer/PDOs is not novel in the field. While this study includes PDO experiments and *in vivo* data, the main message is now different to before in terms of novelty and significance.

We appreciate your comment regarding the existing knowledge on ribosome biogenesis in colorectal cancer and PDOs. While the connection may have been reported, our study aims to contribute to the development of therapeutics involving KRAS inhibitors by utilizing single-cell RNA sequencing and proteome analysis combined with complementary *in vivo* experiments. We believe this integrative approach adds valuable data to the ongoing research in this area.

Overall, I am happy to leave it to the editor to judge the originality of the work and whether the data presented is sufficient to support the findings.

Thank you for your thoughtful evaluation. We appreciate your consideration and will respect the editor's judgment regarding the originality of our work and the sufficiency of the presented data.